# Mixed methods pilot evaluation of a gender-sensitivity training for HIV care providers in Uganda: Effects on providers and clients

Katelyn M. Sileo[1,2]*, Rhoda K. Wanyenze[3], Asha Anecho[3], Rebecca L. Luttinen[4], Katherine Weston[1], Barbara Mukasa[5], Semei C. Mukama[5,6,7], Sten H. Vermund[8], Shari L. Dworkin[9], John F. Dovidio[10], Barbara S. Taylor[11], Trace S. Kershaw[12]

1 Department of Public Health, The University of Texas at San Antonio, San Antonio, Texas, United States of America, 2 Now at: The Department of Discovery and Implementation for the Common Good, Connell School of Nursing, Boston College, Chestnut Hill, Massachusetts, United States of America, 3 Department of Disease Control and Environmental Health, Makerere University School of Public Health, Kampala, Uganda, 4 Department of Sociology and Demography, The University of Texas at San Antonio, San Antonio, Texas, United States of America, 5 Mildmay Uganda, Kampala, Uganda, 6 Infectious Disease Institute, Kampala, Uganda, 7 Mash Research Africa, Kampala, Uganda, 8 Office of the Dean, College of Public Health, University of South Florida, Tampa, Florida, United States of America, 9 School of Nursing and Health Studies, University of Washington, Bothell, Washington, United States of America, 10 Department of Psychology, Yale University, New Haven, Connecticut, United States of America, 11 Department of Medicine, Division of Infectious Diseases, Joe R. & Teresa Lozano Long School of Medicine, UT Health San Antonio, San Antonio, Texas, United States of America, 12 Department of Social and Behavioral Sciences, Yale School of Public Health, New Haven, Connecticut, United States of America

* katelyn.sileo@bc.edu

## Abstract

In sub-Saharan Africa, gender norms shape women's and men's barriers to HIV care engagement, and influence providers' behaviors. Interventions are needed to build providers' capacity for delivering gender-sensitive HIV care. We pilot tested a gender-sensitivity training for HIV care providers and staff in Uganda. Using a quasi-experimental controlled trial (2022–23), we assessed a 4-session intervention, developed by the study team, focused on: gender norms; skills to address HIV care barriers, stigma, and gender-based violence; gender bias recognition/reduction; and client-centered communication. Six clinics were matched (e.g., size, services) and randomly allocated to training intervention or standard-of-care. We enrolled 144 HIV care providers (61 intervention, 83 control) and a cohort of 238 clients with HIV, newly initiated on ART or struggling with adherence (119 per arm). Participants completed structured questionnaires at baseline, 6- and 12-months, and training providers participated in an exit focus group or interview (n = 53). We tested intervention effects using generalized linear models and thematically analyzed qualitative data. Intervention providers reported increased gender-sensitive care competence compared to the control group (Wald $\chi^2 = 20.94$, p < 0.001), supported by qualitative reports of greater gender-related knowledge, perceived importance of gender-sensitive

**Data availability statement:** The quantitative data used for this paper has been made available as supplemental files. The qualitative data has not been made available to protect the participants privacy given difficulty in fully de-identifying the qualitative data for providers working at known clinics with known occupational roles.

**Funding:** This work was supported by the National Institute of Mental Health (K01MH121663 to KS). The funder had no role in study design, data collection and analysis, decision to publish, or preparation of the manuscript. The content is solely the responsibility of the authors and does not necessarily represent the official views of the National Institutes of Health.

**Competing interests:** The authors have declared that no competing interests exist.

care, and skill development. They also reported increased empathy towards clients (B = 0.17, 95% CI = 0.00–0.33, p = 0.04) and use of stress regulation techniques (B = 0.42, 95% CI = 0.13–0.72, p = 0.005). Qualitative data indicated positive effects on client-centered practices (e.g., rapport-building, empathy, eliciting client agendas) and reduced gender bias. No treatment effects were observed in ART adherence or perceived care quality, but clients at intervention clinics reported greater stigma reduction (Wald $\chi^2$ = 18.72, p < 0.001). This study suggests gender-sensitivity training may improve provider practice and reduce stigma, supporting further testing.

## Introduction

In Uganda and elsewhere globally, gender norms are a significant driver of the HIV epidemic, which is primarily driven by heterosexual transmission between cis gender women and men [1]. For women, gender norms reinforcing gender inequity and women's lower status can limit women's autonomy in HIV prevention and treatment decision-making [2,3], create reliance on men for economic survival [4], and drive gender-based violence (GBV), a risk factor for HIV [5]. Consequently, women are at elevated risk for HIV throughout sub-Saharan Africa, particularly adolescent girls and young women [6]. In Uganda, HIV prevalence is 6.5% vs. 3.6% in adult women vs. adult men [7]. For men, masculine norms, or cultural expectations of what it means to be a man, similarly drive men's HIV risk behavior [8], while norms of masculine strength, self-reliance, and respect intersect with HIV stigma to undermine men's engagement across the HIV care continuum [9,10]. Uganda has made progress to close the gap between men and women's engagement in HIV care in recent years; however, continued efforts focused on men's HIV testing and care support are still needed [11]. As such, it is critical that HIV services are designed to be responsive to issues related to gender.

The spectrum of gender responsive health programs range to include services that are gender-sensitive (recognize the different needs of women and men but are not focused on gender inequalities), gender-specific (acknowledge gender norms and consider women's and men's specific needs), and gender-transformative (address the cause of gender-based health inequities and work to transform gender roles, norms, and relations) [12,13]. Strengthening the gender responsiveness of HIV care aligns with the President's Emergency Plan for AIDS Relief (PEPFAR)'s priority to ensure client-centered HIV care [14]. PEPFAR defines client-centered care as services that meet the needs of individuals by increasing convenience, providing welcoming services to diverse populations, making services supportive and responsive, and engaging communities and stakeholders [14]. Considerable work has been done to build the gender-responsiveness of health and community systems in the region, such as the strengthening of gender-sensitive data monitoring systems [15] and the use of gender-specific peer educators and support groups [16,17]. Gender-transformative programs have also gained considerable support in their ability to improve HIV outcomes [18–22].

However, there is a gap in research that focuses on building health care providers' capacity to provide gender-responsive, client-centered HIV care, even though providers are the first line of support in the delivery of antiretroviral treatment (ART) services. Providers are also central to the integration of gender-transformative programs into health systems, which can be complex [23]. For HIV care providers to be responsive to their clients' gender-related issues, they must be aware of and motivated to respond to them. In Uganda, limited training on GBV and low awareness of the prevalence of intimate partner violence among health workers were identified as barriers to the implementation of GBV screening [24,25]. In Malawi and Mozambique, HIV care providers reported low awareness and empathy for men's HIV care engagement barriers [26].

Once aware and motivated to respond, HIV care providers must have acquired the skills needed to respond to clients' gender-related issues. Client-centered communication approaches include tactics to elicit and understand a clients' perspectives and needs within their own unique psychosocial and cultural contexts [27], which could help providers assess and understand gender norm-related barriers to HIV care. Client-centered care delivered through a gender-sensitive lens might also help providers recognize and mitigate power divides and communication issues that are shaped by gender norms. Research shows masculine norms related to respect and autonomy make men especially sensitive to feeling reprimanded or in the position of "learner" relative to health workers [28,29]. Kenyan HIV care providers reported challenges establishing clear roles and sharing power with clients due to conflicting gender versus patient/provider identities [30]. Interventions to strengthen client-centered communication could help mitigate these issues, as it emphasizes the importance of respect, shared decision-making, active listening, and displays of empathy. Research strongly links client-centered care to improved provider communication, linkage and retention in care, adherence, and patient satisfaction [31,32]. Thus, promoting client-centered care within a gender-norms lens may be an effective approach to strengthening gender-sensitive HIV care and improve client outcomes in sub-Saharan Africa.

In addition to being aware, empathetic towards, and able to respond to clients' gendered barriers to HIV care, providers need to be conscious of how their own biases and attitudes might affect their provision of HIV care to not perpetuate gender inequities in client outcomes. One study found health workers held negative bias towards men as HIV care clients; they viewed them as "bad clients" based on assumptions about men being prideful and selfish and expressed little sympathy towards their healthcare engagement challenges [26]. Gender biases held by HIV care providers towards women have also been identified; in a study in Kenya, HIV care providers viewed women as more deceitful about ART adherence, due to HIV status disclosure barriers, and judged their reasons for missing appointments as less legitimate than men's reasons (home vs. work responsibilities) [30]. Culturally-grounded attitudes that unmarried women should not be sexually active have been linked to the stigmatizing treatment of adolescent girls and young women and as a barrier to their receipt of pre-exposure prophylaxis (PrEP) and other HIV services in sub-Saharan Africa [33,34]. Gender biases also intersect and interact with other biases towards people living with HIV, key populations, and other marginalized groups. For example, sex worker stigma is similarly tied to gender norms about women's sexuality [34]. Health workers in Uganda have reported negative attitudes towards transgender and sexual minority populations [35,36]. As such, interventions focused on bias reduction, toward both women and men, should be a core element of gender-sensitive capacity building for HIV care providers.

To increase the quality of HIV care for both women and men, we developed and pilot tested a training program in gender-sensitivity for HIV care providers and clinic staff in central Uganda. The goal of the training was to increase HIV care providers' awareness of how gender norms affect client's HIV care engagement, reduce their gender and related biases, build their client-centered communication skills and their ability to implement gender-specific and gender-transformative approaches to counseling men and women. These intervention goals were focused on cis-gender women and men in rural and peri-urban areas – the main population served by the study clinics. In this paper, we report on the mixed methods findings from a quasi-experimental pilot trial that examined the intervention's preliminary effects on HIV care providers attending the training, as well as on HIV clients receiving care at the clinics where the training was conducted, compared to those at clinics not receiving the training.

Following recommendations for a stage 1b pilot trial (pilot of a new behavioral intervention), our aim was to gather preliminary evidence of trends towards change [37]. We anticipated providers in the training clinics would bolster their skills to provide gender-sensitive care compared to providers in clinics that did not receive the training (standard-of-care). For ART clients, we anticipated those in the training clinics would have increased ART and HIV clinic appointment adherence and explored the intervention's preliminary effects on related indicators of perceived quality of/satisfaction with HIV care and experience of HIV stigma.

## Methods

### Ethics statement

The institutional review boards at the University of Texas at San Antonio and the Makerere University School of Public Health approved the study, as did the Ugandan National Council for Science and Technology. All participants provided written informed consent.

### Design and procedures

To pilot test the gender-sensitivity training, we used a QUAN (+qual) mixed methods embedded experimental design, where the core component is quantitative and the supplemental component is qualitative [38]. In this paper, we present the secondary aim of this trial, which was to explore the preliminary effects of the implementation strategy on relevant HIV care provider and client outcomes. The results of the primary aim of the trial, to establish intervention feasibility and acceptability [37], has been reported separately [39]. The QUAN (+qual) embedded design included a quasi-experimental controlled trial, implemented from 20-2023, to explore preliminary intervention effects by comparing two matched clusters (health facilities) randomly allotted to receive either the gender-sensitivity training intervention or no intervention through 12-months follow-up. The qualitative component, used to interpret and expand on the quantitatively measured intervention effects [38], was comprised of focus groups and in-depth interviews with intervention participants after the 12-month quantitative assessment.The trial was registered with clinicaltrials.gov (NCT05178979) on November 15, 2021, and the protocol has been published [40].

This study was carried out in partnership with Mildmay Uganda, a community-based organization that serves as a Ministry of Health (MOH) implementing partner in the provision of free HIV services in several districts in Uganda. The study took place at Mildmay Uganda-supported governmental health facilities serving both rural and peri-urban communities Luwero, Mityana, and Wakiso Districts. Health facilities were selected in consultation with Mildmay Uganda based on similarities in facility characteristics (e.g., semi-rural setting), number of ART clients, number of HIV care providers/staff, and HIV service delivery following MOH guidelines (immediate ART initiation, routine viral load monitoring). The study was originally designed as a two-site, quasi-experimental pilot trial. Two hospitals — one per arm — were initially selected and randomly allocated via coin toss to receive either the intervention or serve as control. After recruitment began, to meet targets for the HIV client cohort, two smaller clinics were added to each study arm. These additional clinics were selected using the same matching criteria and were located within the same geographic districts as the original sites. They were treated as extensions of the original facilities for implementation purposes. To reduce the risk of cross-contamination, clinics in the intervention and control arms were located more than one hour apart, minimizing the likelihood of clients or providers crossing between sites.

We purposively sampled providers and staff that had regular interaction with HIV care clients at each facility, with the goal of recruiting the entire population. This was inclusive of medical officers, clinical officers, HIV nurses, midwives, linkage facilitators, counselors, lay health workers (e.g., peer mothers, expert clients, male champions, youth/adolescent peer support), and data/recorders officers/assistants. All staff were 18 years of age or older and were fluent in Luganda, the local language. We first sought support for the training and entry/introduction to the clinics through meetings with relevant HIV and gender stakeholders at the MOH-, district-, and facility-levels, during which, we informed them about the study

and elicited their feedback on the intervention. The ART clinic in-charge at each clinic then organized a meeting where a research assistant from our team informed providers/staff about the study. The research assistant obtained providers' written informed consent and conducted the interviewer-administered baseline questionnaire at the time of enrollment. To mask the randomization of intervention and control, research assistants consented all providers for participation in all aspects of the training intervention.

We purposively sampled equal numbers of women and men clients. A research assistant reviewed clinic records to identify potentially eligible participants. Clinic staff then referred these clients to the research assistant via phone call or when they presented to the clinic. The research assistant informed them of the study, assessed eligibility with a computerized screening tool, obtained written informed consent, and conducted the interviewer-administered baseline questionnaire in the clinic or another agreed upon private location at the time of enrollment. The use of clear inclusion/exclusion criteria programmed into a computerized interviewer-administered questionnaire was used to reduce selection bias. For clients to be eligible, they had to be enrolled in HIV care at one of the participating facilities, 18 years of age or older (or an emancipated minor who are legal adults in Uganda based on being married and/or having children), and fluent in Luganda. They also had to be either pre-ART (newly diagnosed), newly initiated on ART (within one year), or struggling with treatment adherence, defined in two ways: virally unsuppressed based on most recent viral load results within six months, obtained through clinic records, or self-reported ART non-adherent (any missed doses) in the prior two weeks. Through these inclusion criteria, we sought to engage priority groups for ART counseling and ensure that we would be able to see changes over time in ART adherence and related outcomes.

## Sample size justification

As a pilot trial with an exploratory aim of examining our intervention's preliminary effects, our sample size is based on guidelines for Stage 1b studies (pilot testing of new behavioral interventions), which suggest 15–30 participants per condition [37]. We aimed to recruit the full population of HIV care providers/staff per clinic, which was estimated to be ~20–35 providers per clinic and fit within these guidelines. We also aimed to meet these standards for the client cohort, but since the intervention effect would be indirect on clients, we established a larger a target sample of 120 client participants per treatment arm *a priori* by exploring the study's power to detect change in clients' ART adherence (the main client cohort outcome). We used G*Power (v. 3.1) [41] (assuming power = .80, α = .05) to confirm a sample of 100 patients per condition (estimating 17% attrition from n = 120) would detect a moderate effect size (d = 0.4) in ART adherence, based on ART adherence interventions from sub-Saharan African studies [42]. For the main qualitative portion of the study, the aim was to engage the entire intervention arm's sample in a follow-up assessment.

## Description of study arms

**Control.** In the control clinics, the providers received no intervention; thus, the intervention was compared to the standard-of-care implementation of MOH ART guidelines [43]. After the study ended, we provided the training to the control clinics to ensure equal benefit to participating clinics.

**Intervention.** The intervention was initially informed by existing strategies developed by Dovidio and colleagues [44], based on research that has shown that increasing provider motivation, awareness, skills, empathy, and emotional regulation can prevent implicit racial and gender biases from affecting clinical judgment and behavior. For the current intervention, these strategies were adapted to increase providers' knowledge, motivation, skills, and empathy to equitably deliver Ugandan MOH ART program guidelines [43] to men and women clients. In our adaption of these guidelines, we sought to increase awareness of HIV gender disparities, increase empathy/skills to counsel men and women's gendered barriers to care, promote collaborative decision-making with clients, and promote client-centered and gender-sensitive/specific communication in ART and other HIV-related counseling. The development of content based on these strategies was informed by: 1) the existing literature from sub-Saharan Africa on gender norms' influence on client engagement

in HIV care and on the provision of quality HIV care (focused on the provider-client relationship); and 2) a local needs assessment that was carried out in ART clinics and local communities as a pre-cursor to the trial (inclusive of qualitative data collected from HIV care providers, clients, community members, and HIV care stakeholders); 3) several rounds of direct feedback on the approach and materials from HIV care stakeholders from Mildmay Uganda, the Uganda MOH, and local District Health Teams.

The final intervention package (outlined in Table 1) included four training sessions facilitated by two national trainers experienced in the capacity building of health workers in topics related to HIV and gender. There were a total of 3 training cohorts in the intervention arm (participants' cohort aligned with their clinic). Sessions 1–3 included directive education, facilitated group discussions, interactive role plays, and vignettes. Session 4 was a practicum-based session that reviewed the content delivered in sessions 1–3. Session content was delivered in Luganda and English (tailored to the needs of the group) and participants received a training workbook to follow along and take notes in.

Sessions 1 and 2 were delivered over two consecutive days in an off-site setting (outside of the health facility). As outlined in Table 1, Session 1 content was comprised of an introduction to gender (e.g., sex vs. gender, gender norms/roles), and gender's relationship to: HIV acquisition, barriers to HIV care engagement, and HIV gender disparities—content was grounded in examples specific to the local context. We introduced participants to the spectrum of gender transformative health programs [12,13], with examples applied to HIV services. We engaged participants in an interactive, role play-based workshop that explored how HIV can be viewed as threatening to core elements of masculinity/femininity, leading to HIV stigma, which we referred to as the "HIV threat framework" and created based on prior work from Naugle et al. [45] and the broader literature on gender norms and HIV [9,10]. Through interactive roleplays, providers were taught gender-sensitive, gender-specific, and gender-transformative ways to identify and reduce the "HIV threat" for clients, and thus, mitigate HIV stigma. For example, facilitators encouraged providing referrals to services to reduce HIV stigma (e.g., gender-sensitive/specific support groups) and modeled ways to counsel clients on ART's ability to restore/maintain their gender roles (gender-specific) and to reject gender norms/roles that conflict with HIV care engagement (gender-transformative).

Session 2 included an introduction to bias and stereotypes, with discussion-based explorations of locally relevant stereotypes and biases related to gender, HIV and other marginalized groups that may negatively affect the provision of quality, equitable HIV care. Through an adaption Bower et al.'s [46] "three-perspective framework," facilitators taught providers to assess clients' needs through the client perspective (e.g., personal barriers to care, perceptions of illness), their own perspective (including reflection on one's own biases), and the societal perspective (incorporating the broader social/cultural context, focusing on gender norms). Facilitators led individual and group exercises that were adapted from Bower et al. [46], applying this framework to a "critical incident" or a real, past experience with a "difficult" client (including both a man and woman client), with the goal of helping providers recognize how their own biases and emotions may negatively affect their provision of HIV care, and to increase empathy for client challenges, including those related to gender. Facilitators presented evidence on how external stressors and stressful work environments can increase the likelihood that bias enters clinical interactions, and shared how to deploy evidence-based emotional and stress regulation techniques (e.g., breathing exercises) to mitigate bias and stress from affecting client relationships [44].

The other modules in session 2 covered how gender intersects with power dynamics in provider-client relationships, and shapes men and women's communication styles, with guided discussions on how gendered power in provider-client dyads might impact providers' relationships with clients. Client-centered care/communication was introduced as an approach to providing gender-sensitive care to clients, helping to overcome power imbalances and improve verbal/non-verbal communication with clients, and providing a framework to elicit and respond to gender preferences for care and gendered barriers to HIV care. Through participatory role playing specific to gendered barriers to HIV care including "HIV threats" (i.e., HIV stigma viewed through a gender lens), participants practiced the application of client-centered techniques for building rapport, eliciting the clients' agenda, verbal and non-verbal communication skills, expressing empathy, and engaging in collaborative decision-making, using content adapted from Hofert et al. [47].

**Table 1. Overview of content per session in a gender-sensitivity training for HIV care providers and clinic staff that interact with HIV clients, Uganda 2021-2023.**

| Session, Location, Duration | Session Content |
|---|---|
| **Session 1**<br>Full day, off-site | **Module 1: What is Gender? Gender & HIV** |
| | • Understanding gender vs. sex, gender norms and roles, societal gender inequity, and their link to HIV outcomes and disparities |
| | **Module 2: Reducing the "HIV threat"**<br>• Introduction to the range of gender focus in health programs (gender inequitable, gender neutral, gender sensitive, gender specific, and gender transformative health programs) with examples applied to HIV care<br>• The link between HIV stigma and gender norms and recommendations in the application of gender-sensitive, gender-specific, and gender-transformative approaches to respond to HIV stigma and gendered barriers to HIV care engagement and inequities, and gender disparities in the HIV care continuum in Uganda |
| **Session 2**<br>Full day, off-site | **Module 3: What is bias?** |
| | • Introduction to bias and stereotypes as part of the normal human thought processes<br>• Exploration of gender, HIV, and other biases common in the local context<br>• Client perspective exercise to encourage providers to recognize: 1) clients' gendered barriers to care and increase empathy for client challenges and 2) how their own biases may negatively affect their provision of HIV care<br>• Introduction to emotional/stress regulation techniques that can be used to prevent bias/stress from affecting relationships with clients |
| | **Module 4: Communication and Power dynamics** |
| | • Exploration of gender and other power dynamics that shape provider/client relationships<br>• Identification of socialized gendered communication styles in Uganda and their effect on provider/client relationships and the provision of quality HIV care |
| | **Module 5: Client-Centered Care and Communications Skills** |
| | • Introduction to client-centered care and communication as central to providing gender-sensitive care<br>• Recommendations and techniques in building rapport, eliciting the patients' agenda, verbal and non-verbal communication skills, expressing empathy, and engaging in collaborative decision-making<br>• Application of client-centered counseling to help clients overcome gendered barriers to HIV care engagement |
| **Session 3**<br>2 hours, on-site | **Module 6: Gender-Based Violence (GBV) and HIV care\*** |
| | • What is GBV? Links to gender norms, health effects, and intersection with HIV<br>• Roles of HIV providers/staff in responding to GBV |
| | \* Content was a refresher of the Ministry of Health (MOH)'s existing GBV training for healthcare providers, enhanced to connect GBV to gender norms and HIV risk and care engagement, and to frame responsiveness to GBV as an important aspect of providing gender-sensitive care, client-centered care. |
| **Session 4**<br>2 hours, on-site | **Applied Refresher Training** |
| | • Facilitators (Expert Trainers) observed one-on-one provider-client interactions and provided individualized feedback on their gender-sensitive and client-centered care using a checklist<br>• After observations and individualized feedback, facilitators met in small groups with providers to lead a brief refresher of the training's content, and a guided, discussion to group problem-solve challenges in implementing the training's content in practice |

Notes: Session 1 and 2 were delivered in two consecutive days in an off-site training. Session 3 was approximately 1–2 weeks later delivered on-site (at the facility). Session 4 was delivered approximately 3 months later, on-site. Sessions 1–3 were delivered by two Expert Trainers (medical professionals experienced in the delivery of HIV training and mentorship for capacity building). Session 4 was delivered by one Expert Trainer. Sessions 1–3 included a mix of directive education and active learning techniques (interactive discussion, role-plays, vignettes).

Session 3 consisted of a 2-hour on-site group training session conducted at the clinics. The content focused on gender-based violence (GBV), developed to reinforce the MOH's existing GBV training, highlighting GBV's effects on health and the health worker's role in responding to GBV [48]. We built on this content to reinforce and not duplicate existing MOH GBV content but enhanced it by more explicitly linking between GBV to HIV (the MOH training is for all health workers) and by linking GBV to session 1 and 2's content on broader gender norms and inequity. Facilitators emphasized the importance of responding to GBV as part of providing gender-sensitive, client-centered care.

The final session was a "refresher" or review session of sessions 1–3, with a focus on the practical application of the content into care. The session began with one of the facilitators directly observing providers' interaction with a client in the ART clinic. The facilitator made notes on the interaction and completed a checklist that included a list of client-centered communication and gender-sensitive counseling techniques; the checklist was adapted from an existing client-centered communication assessment for providers [47]. The checklist was piloted before use and adapted with feedback from the facilitators. One facilitator observed up to four providers on a given day. After the observation, the facilitator provided individual feedback to the providers based on the observed interaction, and then met with the small group of providers. In the group discussion, the facilitator provided a review of core session content following a structured guide, and then facilitated a group discussion following semi-structured prompts on successes and challenges in translating the training's content into practice. The facilitator provided suggestions for overcoming the identified challenges, and encouraged group-problem solving, eliciting other providers' suggestions based on their own experiences.

## Data collection procedures and measures

**Quantitative.** Experienced interviewers conducted a computerized, structured interview following enrollment and again at approximately 6- and 12-months follow-up for participants in both the HIV care provider/staff and client cohorts. Interviews took place in a private setting, including the clinic or another agreed upon location, or over the phone, if necessary. Participants in both cohorts received 10,000 Ugandan Shillings (UGX) (~$3) per baseline and 6-month assessment and 18,000 UGX (~5 USD) for the 12-month assessment, aligned with the ethics review board standards. At baseline for providers, socio-demographics, occupational role, and years working at the current facility and with HIV clients were collected. For HIV clients, socio-demographics were also collected at baseline, along with health and HIV history. As detailed in Table 2, measures to examine change in the intervention's hypothesized primary and secondary outcomes for the HIV provider and client cohorts were collected, along with additional validated measures for potential covariates relevant for HIV care engagement outcomes.

**Qualitative.** All participants who attended the training across the three intervention sites were invited to participate in an exit focus group discussion that took place after the 12-month quantitative assessment. Focus groups were designed to be homogenous by type of health care role and each group included 4–8 people, aligned with best practices in qualitative research in focus groups [49]. A total of 46 providers attended 1 of 7 focus groups, which were conducted at each clinic and segregated by cadre (4 health worker groups and 3 lay health worker groups); 7 additional participants who were not available to attend a focus group discussion participated instead in an individual interview. Taken together, 87% of intervention arm providers completed a qualitative follow-up. A trained and experienced qualitative interviewer led the focus groups and interviews following an interview guide with questions that elicited participant feedback on their experience and perceptions of the training sessions, including questions on what they learned and what changes they made (if any) to their practice. An experienced qualitative interviewer, independent of the study team, moderated and transcribed the focus group discussions. The interviewer was trained on the strengths (capitalizing on group dynamics) and weaknesses (views produced at the poles of debates) of focus groups and were trained to probe neutrally to ensure that a range of views were represented. Focus groups lasted approximately 90 minutes, and individual interviews lasted approximately 30 minutes. Participants received 25,000 UGX (~7 USD) for their participation. All sessions were conducted in Luganda and audio recorded by a trained research assistant; transcripts were translated into English.

## Data analysis approach

First, we used generalized linear models [50] in SPSS v. 28 [51] to examine baseline equivalence between arms in both HIV care provider and client cohorts, controlling for clinic. We also used generalized linear models to evaluate the time by intervention effect on primary and secondary outcomes. We included clinic as a fixed effect to account for potential between-cluster variation in outcome measures. Given the limited number of clusters in this pilot trial, a

Global Public Health

**Table 2. Quantitative measures, gender-sensitivity provider training intervention pilot trial, Uganda 2021–2023.**

| Provider cohort | |
|---|---|
| **Outcomes** | **Measures** |
| *Competence for gender-sensitive care* | Competence for gender-sensitive care was measured through an adaption of two existing scales to measure awareness, attitudes towards, and self-efficacy to provide gender-sensitive care. The first scale was Saha et al.'s Self-Rated Cultural Competence Instrument for Primary Care Providers that assesses awareness, perceived importance, motivation, and skills to provide culturally competent care [48]. The second scale was the Adapted Nijmegen Gender Awareness in Medicine Scale (N-GAMS) [49], developed for medical personnel. For the current study, all items were adapted to be specific to competence for gender-sensitive HIV care. Cronbach's α at baseline in current sample = 0.67. |
| *Communication Self-Efficacy* | An adapted version of the Self-Efficacy Questionnaire (SE-12) for Provider Communication was used to assess communication self-efficacy, adapted to be gender-specific. Cronbach's α at baseline in current sample = 0.75. |
| *Empathy* | Provider empathy for clients' experiences was measured from an adapted version of the Jefferson Scale of Physician Empathy, which has been adapted for HIV care previously [50,51]. Cronbach's α at baseline in current sample = 0.58. |
| *Emotional regulation and stress reduction techniques* | Providers' use of emotional regulation and stress reduction techniques, such as breathing exercises, sense soothing, tension release, attention shifting, and positive reframing, were measured through items adapted from the Mindful Self-Care Scale (MSCS) and the Brief COPE [52,53] and align with techniques taught in the training. Cronbach's α at baseline in current sample = 0.71. |
| Client cohort | |
| **Outcomes** | **Measures** |
| *ART adherence* | Measured by self-report through the Adult AIDS Clinical Trials Group (AACTG) scale's [54] 4-day adherence recall questions; demonstrated good construct validity in Uganda [55], strong correlations with viral load [56], and moderate correlations with electronic adherence monitoring [57]. For this study, adherence is operationalized as the proportion of pills taken as prescribed in the 4-day recall period. |
| *HIV appointment visit adherence* | Proportion of kept visits/scheduled visits (kept + missed visits) (continuous measure, range = 0.0–1.0), collected through client clinic records and triangulated with participant self-report. If the participant stated that the clinic records were wrong or missing, self-report was recorded. |
| *Quality of communication* | Client's perceptions of the quality of communication with their HIV care providers was measured through two scales that were combined for a total score continuous score. Wilson et al. [58] was developed for HIV populations, including items measuring the perceived quality of general health communication from HIV care providers, asking patients to rate the quality of their HIV care providers in communicating general health information and in providing HIV specific information. In addition, clients' perceived quality of provider communication specific to ART adherence was be measured from items adapted from Schneider and colleagues [59]. Cronbach's α at baseline in current sample = 0.89. |
| *Participatory decision-making* | Participatory decision-making style of HIV care providers, or how active of a role clients perceive they have in their health care decisions, was measured with Kaplan's 7-item scale [60]. Cronbach's α at baseline in current sample = 0.77. |
| *Overall satisfaction with care* | The GHAA Consumer Satisfaction Survey measures overall satisfaction with care [61]; we adapted these items to focus specifically on HIV care to measure client satisfaction with HIV care. Cronbach's α at baseline in current sample = 0.75. |
| *HIV stigma* | HIV stigma was measured using Earnshaw's HIV stigma framework scale [61], which measures anticipated (the degree of HIV stigma[62] one expects to encounter from others), enacted (the degree of HIV stigma one has encountered from others), and internalized HIV stigma (the degree to which one internalized negative beliefs about people living with HIV about themselves). Cronbach's α at baseline in current sample = 0.91. |

random effects approach would not provide stable estimates; thus, treating clinic as a fixed effect allows for more reliable control of cluster-level confounding [52,53]. We explored baseline differences between arms and included variables that differed at a level of $p < 0.10$ as covariates in preliminary models testing intervention effects. This approach was used to identify potential imbalances and inform model specification in this exploratory, pilot context. Final models retained covariates based on their theoretical relevance, observed association with outcomes, and/or meaningful influence on the intervention effect estimate, rather than solely on statistical significance. We report the time by arm interaction. For all provider and client analyses, we tested the time by intervention by gender effect. For providers, we also

tested the time by intervention by cadre effect, categorizing providers into certified health workers (licensed doctors, nurses, counselors) vs. lay health workers or non-specialized staff (e.g., peers). These interactions are only presented if they were statistically significant ($p < 0.05$). We report unstandardized betas (B) with 95% confidence intervals (CIs) for continuous outcomes and odds ratios (ORs) and adjusted odds ratios (AORs) with 95% confidence intervals (CIs) for binary outcomes.

The qualitative data were analyzed thematically [54]. To explore the specific question of the intervention's potential effects or null effects on HIV care provider and client outcomes, we developed the coding guide *a priori* based on the intervention's hypothesized conceptual model (e.g., effect on providers' knowledge, attitudes, behavioral skills, and behavior), which was modified through iterative review of the transcripts. A team of three trained research assistants (a Ugandan bachelors-level research assistant, a U.S. Ph.D. student, and a U.S. bachelor's student, [AA, RLL, KW]) used an iterative process to apply codes to transcripts, while writing initial impression summaries for each transcript to help inform the coding guide. Coders met weekly with the project director (KMS) to discuss new codes and potential themes. The team resolved discrepancies through discussion and consensus. New codes were drawn inductively, including sub-codes within the overall intervention effect domains. Codes were defined in a shared codebook, and decision trails were kept to ensure accuracy and consistency in the application of codes. The team coded directly in the transcripts first and then organized codes into an Excel spreadsheet. KMS collated codes that represented thematic elements and summarized themes with representative quotations, with support from the coders. The team identified final themes through review, discussion, and consensus, with input from the broader investigative team and community stakeholders, who were presented the findings during community dissemination events.

To mix the quantitative and qualitative data, following the QUAN (+qual) design, we analyzed the quantitative data, while qualitative coding was ongoing, followed by the thematic analysis of the coded qualitative data. We compared the qualitative results with the quantitative results on intervention effects at the point of the presentation and interpretation of results. We present them together in the results narrative, organized by the overarching thematic areas explored following the intervention's conceptual model. We used the qualitative data to confirm and expand on intervention effects supported by the quantitative data and to provide insight into intervention elements that had an effect or to explain null effects. The two data sources also served to validate each other.

## Statements and declarations

The authors have no competing interests to disclose.

## Results

Fig 1 displays the Consolidated Standards of Reporting Trials (CONSORT) study diagram. Of the HIV care providers/staff screened for participation, the final enrolled samples in each study arm were 61 in the intervention clinics and 83 in the control clinics; only 2 eligible providers (3.2%) declined participation in the intervention clinics and no eligible providers declined in the control clinics. At 6- and 12-month follow-up, 98% (n = 60/61) of providers completed the 6- and 12-month questionnaires in the intervention clinics, compared to 94% (n = 78/83) and 96% (n = 80/83) in the control clinics. Provider attendance of the intervention sessions was 86%. Of all planned content, 100% of intervention steps were delivered; 78% were delivered with fidelity or as intended by the study investigators, measured through review and assessment of a random selection of 20% of recorded session transcripts.

For the client cohort, a total of 119 individuals per study arm were recruited at baseline. In the intervention clinics, 81% (97/119) completed the 6-month assessment and 88% (105/119) completed the 12-month assessment. In the control clinics, 76% (n = 91/119) completed the 6-month assessment and 90% (108/119) completed the 12-month assessment.

Table 3 details the sample characteristics of the HIV care provider/staff cohort at baseline (N = 144). There were more women (72.2%) than men (27.8%)— an expected imbalance, as there are more women health workers than men

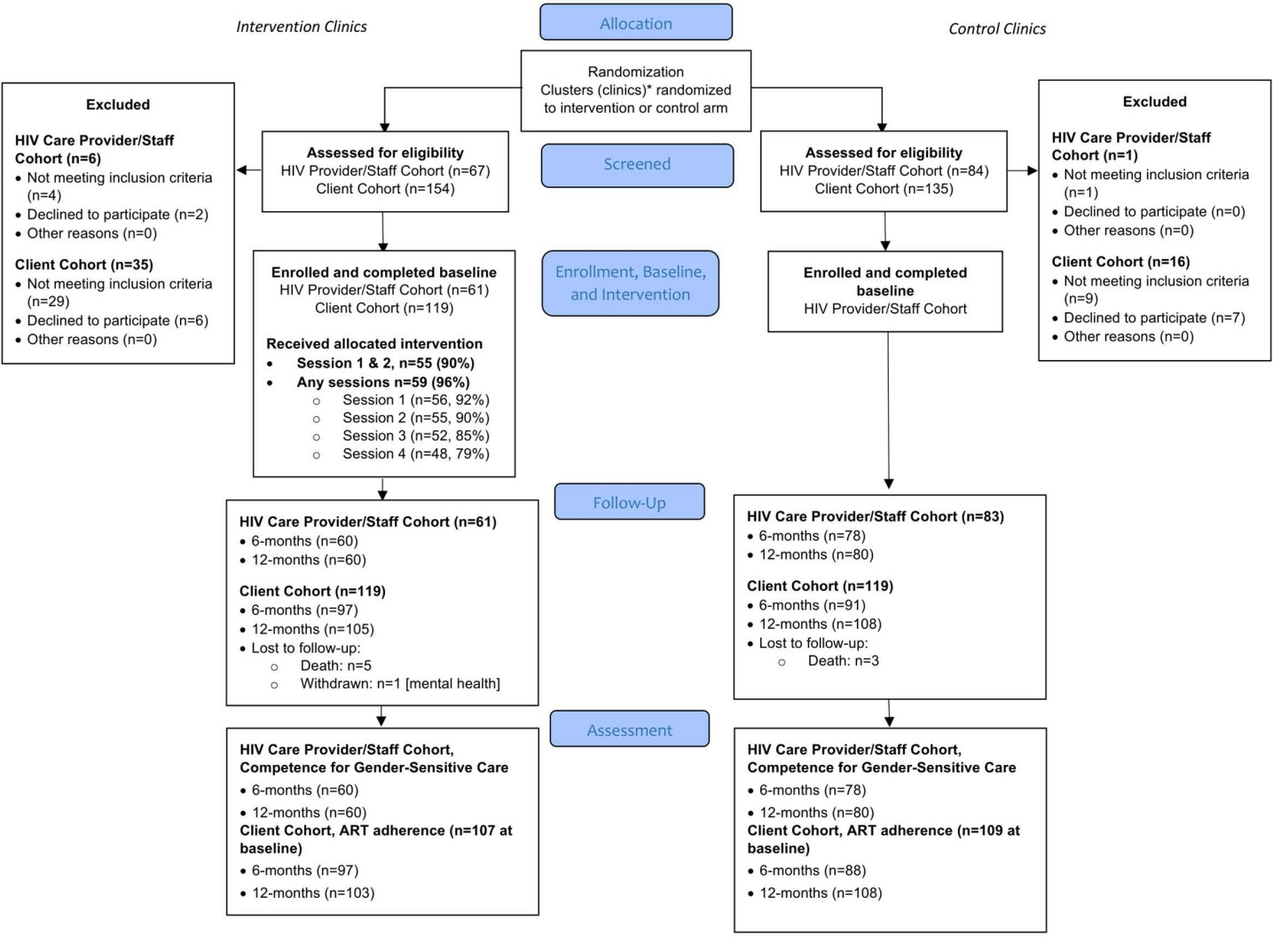

**Fig 1. CONSORT diagram.**

in Ugandan health facilities. The full range of provider occupations are detailed in Table 3; 62.5% were considered lay health workers or other non-clinical staff with regular interaction with HIV clients vs. 37.5% certified health workers or counselors. Providers had been working at their specific health facility an average of 4.7 years (standard deviation [SD]=5.3) and practicing in their current role an average of 7.1 years (SD = 7.5). No statistically significant differences were identified between arms on these or other baseline characteristics, but providers in the control clinics were in their current clinic and role slightly longer than those in the intervention clinics and had a slightly higher percentage of lay staff (see Table 3).

The client cohort (Table 3) was balanced by sex in both arms, based on purposive, equal recruitment of men and women. The average age of the overall sample was 39.2 (SD = 11.3), and the majority were from the Buganda tribe (66.0%) with primary education or less (69.7%). Most participants were already on ART at baseline (92.4%), among which, the average number of years on ART was 4.4 (SD = 4.9). The analysis for baseline equivalence identified several characteristics where the two arms differed at baseline (p < 0.05). Details are in Table 3, but in summary, clients from

**Table 3. Sample characteristics at enrollment by study arm for the HIV care provider/staff and client cohorts, with comparisons between study arms, gender-sensitivity training intervention pilot trial, Uganda 2021–2023.**

| | Full Sample (N = 144) | Intervention (n = 61) | Control (n = 83) | Comparison between arms B (95% CI)/ OR (95% CI) | p |
|---|---|---|---|---|---|
| **HIV Provider/Staff Cohort** | | | | | |
| Gender | | | | | |
| Women | 104 (72.2%) | 45 (73.8%) | 59 (71.1%) | 1.60 (0.56, 4.59) | 0.38 |
| Men (ref) | 40 (27.8%) | 16 (26.2%) | 24 (28.9%) | | |
| Age | 33.6 (10.8) (range: 19–69) | 31.8 (10.8) (range: 19–63) | 34.9 (10.7) (range 19–69) | -6.88 (-11.44, -2.31) | **0.003** |
| Education | | | | | |
| University/Nursing/Medical/ Graduate School | 54 (37.5%) | 23 (37.7%) | 31 (37.3%) | 1.40 (0.54, 3.60) | 0.49 |
| Primary/Secondary/Vocational/ Certificate (ref) | 90 (62.5%) | 38 (62.3%) | 52 (62.7%) | | |
| Occupation (all categories) | | | | | |
| Clinical Officer | 9 (6.3%) | 3 (4.9%) | 6 (7.2%) | | |
| Nurse | 27 (18.8%) | 13 (21.3%) | 14 (16.9%) | | |
| Counselor | 18 (12.5%) | 9 (14.8%) | 9 (10.8%) | | |
| Peer Educator | 34 (23.6%) | 20 (32.8%) | 14 (16.9%) | | |
| Linkage Facilitator | 15 (10.4%) | 5 (8.2%) | 10 (12.0%) | | |
| Laboratory Technician | 9 (6.3%) | 2 (3.3%) | 7 (8.4%) | | |
| Village Health Team (VHT) | 16 (11.1%) | 3 (4.9%) | 13 (15.7%) | | |
| Records/Data Officer/Assistant | 13 (9.0%) | 3 (4.9%) | 10 (12.0%) | | |
| Screener | 3 (2.1%) | 3 (4.9%) | 0 (0.0%) | | |
| Cadres categorized | | | | | |
| Certified health workers and counselors | 54 (37.5%) | 25 (41.0%) | 29 (34.9%) | 0.96 (0.37, 2.51) | 0.94 |
| Lay health workers or other clinic staff (ref) | 90 (62.5%) | 36 (59.0%) | 54 (65.1%) | | |
| Number of years working with HIV clients | | | | | |
| Less than 5 years | 75 (52.1%) | 30 (49.2%) | 45 (54.2%) | 0.90 (0.36, 2.27) | 0.83 |
| 5 years or more | 69 (47.9%) | 31 (50.8%) | 38 (45.8%) | | |
| Number of years practicing in current role | 7.1 (7.5) (range: 0–35) | 6.5 (7.05) (range: 0–26) | 7.6 (7.8) (range: 0–35) | -3.89 (-7.13, -0.67) | **0.02** |
| Number of years at current health facility | 4.7 (5.3) (range: 0–26) | 4.2 (5.5) (range: 0–24) | 5.0 (5.1) (range: 0–26) | -2.71 (-4.96. -0.46) | **0.02** |
| | **Full Sample (N = 238)** | **Intervention (n = 119)** | **Control (n = 119)** | **Comparison between arms B (95% CI) / OR (95% CI)** | **p** |
| **HIV Client Cohort** | | | | | |
| | M (SD) or n (%) | M (SD) or n (%) | M (SD) or n (%) | | |
| Gender | | | | | |
| Women | 119 (50%) | 60 (50.4%) | 59 (49.6%) | – | – |
| Men (ref) | 119 (50%) | 59 (49.6%) | 60 (50.4%) | | |
| Age | 39.22 (11.34) range: 18–72 | 38.36 (11.66) range: 18–67 | 39.58 (11.05) range: 20-72 | -3.36 (-7.24, 0.53) | 0.09 |
| Tribe | | | | | |
| Muganda | 157 (66.0%) | 80 (67.2%) | 77 (64.7%) | 1.15 (0.55, 2.39) | 0.71 |
| All other (ref) | 81 (34.0%) | 39 (32.8%) | 42 (35.3%) | | |

*(Continued)*

**Table 3.** (Continued)

|  | Full Sample (N = 144) | Intervention (n = 61) | Control (n = 83) | Comparison between arms B (95% CI)/ OR (95% CI) | p |
|---|---|---|---|---|---|
| Education |  |  |  |  |  |
| Secondary or greater | 72 (30.3%) | 39 (32.8%) | 33 (27.7%) | 2.16 (0.99, 4.71) | 0.05 |
| No Schooling or Primary (ref) | 166 (69.7%) | 80 (67.2%) | 86 (72.3%) |  |  |
| Relationship status |  |  |  |  |  |
| Married or in a steady relationship | 110 (46.2%) | 53 (44.5%) | 57 (47.9%) | 2.00 (0.96, 4.19) | 0.07 |
| Not married or in a relationship (ref) | 128 (53.8%) | 66 (55.5%) | 62 (52.1%) |  |  |
| Monthly income (Uganda Shillings [UGX]) | 158,588.30 (266,873.56) range: 0–2,000,000 | 223,428.57 (324,702.01) range: 0–2,000,000 | 93,748.02 (170,690.90) range: 0–1,000,000 | 2.63 (0.74, 4.51) | **0.006** |
| Years since HIV diagnosis | 4.75 (6.15) range: < 1–34 | 3.71 (5.56) range: < 1–34 | 5.78 (6.56) range: < 1–32 | -2.70 (-4.83, -0.56) | **0.01** |
| On ART |  |  |  |  |  |
| Yes | 220 (92.4%) | 109 (91.6%) | 111 (93.3%) | 0.77 (0.28, 2.18) | 0.61 |
| No | 18 (7.6%) | 10 (8.4%) | 8 (6.7%) |  |  |
| Months since HIV diagnosis, for those not on ART (n = 18) | 3.33 (9.07) Range: 0–28 | 2.00 (6.32) range: 0–20 | 5.00 (10.04) range: 0–28 | – | – |
| Years since ART initiation, for those on ART | 4.38 (4.93) range: < 1–32 | 3.47 (4.09) range: < 1–15 | 5.28 (5.49) range: < 1–32 | -1.88 (-3.77, 0.02) | 0.05 |
| Travel time to clinic | 50.58 (44.08) range: 2–240 | 39.91 (34.36) range: 2–180 | 61.25 (49.92) range: 3–240 | -0.91 (-1.32, -0.51) | **<0.001** |
| Social support | 2.07 (0.85) | 2.01 (0.84) | 2.13 (0.85) | 0.26 (-0.02, 0.55) | 0.07 |
| Any hazardous alcohol use (AUDIT-C) |  |  |  |  |  |
| Yes | 46 (19.3%) | 29 (24.4%) | 17 (14.3%) | 2.64 (1.08, 6.46) | **0.03** |
| No (ref) | 192 (80.7%) | 90 (75.6%) | 102 (85.7%) |  |  |
| Food insecurity | 0.82 (0.83) | 0.96 (0.83) | 0.69 (0.81) | 0.07 (-0.22, 0.36) | 0.64 |
| Depression (CESD score) | 6.32 (4.31) | 6.31 (3.99) | 6.32 (4.65) | -1.27 (-0.86, 0.31) | 0.12 |
| Lifetime experience of any intimate partner violence |  |  |  |  |  |
| Yes | 91 (43.3%) | 61 (64.2%) | 30 (26.1%) | 2.81 (1.28, 6.16) | **0.01** |
| No (ref) | 119 (56.7%) | 34 (35.8%) | 85 (73.9%) |  |  |

Abbreviations: M = mean, SD = standard deviation, B = unstandardized beta, OR=odds ratio, CI = confidence interval.Notes: For reference, 158,588 UGX is approximately 40 USD; for income in the comparison of arms, 1 unit = 40,000 UGX or approximately 10 USD; for travel time to clinic in the comparison of arms, 1 unit = 30 minutes; Bold indicates statistical significance at p < 0.05

intervention clinics, on average, knew their status for fewer years (B=-2.70, 95% CI=-4.83, -0.56, p=0.01) and were on ART for fewer years than those from control clinics (B=-1.88, 95% CI=-3.77, 0.02, p=0.05). Intervention clients reported more hazardous alcohol use (OR = 2.63, 95% CI=0.74, 4.51, p=0.006), food insecurity (B=0.07, 95% CI=0.22, 0.36, p=0.64), lifetime experience of intimate partner violence (OR = 2.81, 95% CI=1.28, 6.16, p=0.01), higher monthly income in units of 40,000 UGX (~10 USD) (B=2.63, 95% CI=0.74, 4.51, p=0.006), and less travel time to the HIV clinic than the control in units of 30 minutes (B=-0.91, 95% CI=-1.32, 0.51, p<0.001).

Table 4 displays the results of the models testing the intervention's effect on each outcome, as well as the descriptive statistics for each outcome at all time points. Table 5 provides an outline of the main qualitative findings with illustrative quotations. In the narrative below, these findings are presented together, organized by overarching thematic areas on the explored intervention effects.

**Table 4. Exploratory outcomes at baseline, 6-month, and 12-month follow-up by condition (unadjusted) and the arm-by-intervention effect (adjusted) for the HIV provider/staff cohort and client cohort, gender-sensitivity provider training intervention pilot trial, Uganda 2021–2023.**

| | Intervention | Control | Intervention Effect by Time Point Arm*Time | | Overall Intervention Effect Arm*Time | |
|---|---|---|---|---|---|---|
| | | | B (95% CI) | p | Wald χ² | p |
| **HIV Provider/Staff Cohort** | | | | | | |
| **Exploratory primary outcome** | | | | | | |
| Competence for gender-sensitive care (overall score) | | | | | 20.94 | **<0.001** |
| 12-months | 3.82 (0.31) | 3.61 (0.31) | -0.02 (-0.14, 0.10) | 0.80 | | |
| 6-months | 3.95 (0.35) | 3.48 (0.25) | 0.24 (0.12, 0.37) | **<0.001** | | |
| Baseline (ref) | 3.78 (0.40) | 3.55 (0.24) | | | | |
| **Exploratory secondary outcomes** | | | | | | |
| Self-efficacy for client-centered communication | | | | | 7.84 | **0.02** |
| 12-months | 2.98 (0.53) | 3.28 (0.49) | -0.22 (-0.44, -0.00) | 0.05 | | |
| 6-months | 3.24 (0.54) | 3.26 (0.43) | 0.04 (-0.18, 0.27) | 0.70 | | |
| Baseline (ref) | 3.08 (0.75) | 3.15 (0.40) | | | | |
| Empathy | | | | | 4.26 | 0.12 |
| 12-months | 3.91 (0.30) | 3.63 (0.44) | 0.13 (-0.04, 0.30) | 0.14 | | |
| 6-months | 3.86 (0.45) | 3.54 (0.32) | 0.17 (0.00, 0.33) | **0.04** | | |
| Baseline (ref) | 3.77 (0.38) | 3.62 (0.38) | | | | |
| Emotional regulation and stress reduction techniques | | | | | 8.17 | **0.01** |
| 12-months | 2.03 (0.97) | 2.06 (0.81) | 0.30 (-0.01, 0.61) | 0.05 | | |
| 6-months | 1.93 (0.80) | 1.86 (0.75) | 0.42 (0.13, 0.72) | **0.005** | | |
| Baseline (ref) | 1.64 (0.91) | 1.99 (0.62) | | | | |
| **Client Cohort** | | | | | | |
| **Exploratory primary outcomes** | | | | | | |
| ART adherence | | | | | 2.53 | 0.28 |
| 12-months | 95.45 (19.51) | 100.00 (0.00) | -3.05 (-9.21, 3.11) | 0.33 | | |
| 6-months | 95.64 (14.35) | 96.73 (13.33) | 0.39 (-6.67, 7.45) | 0.91 | | |
| Baseline (ref) | 90.14 (19.51) | 91.68 (22.40) | | | | |
| **Exploratory secondary outcomes** | | | | | | |
| Retention in care | | | | | 1.68 | 0.43 |
| 12-months | 0.99 (0.07) | 1.00 (0.00) | -0.02 (-0.05, 0.02) | 0.32 | | |
| 6-months | 0.99 (0.06) | 0.98 (0.08) | -0.01 (-0.04, 0.03) | 0.86 | | |
| Baseline (ref) | 0.96 (0.11) | 0.95 (0.12) | | | | |
| Quality of communication | | | | | 54.80 | **<0.001** |
| 12-months | 2.14 (0.87) | 2.88 (0.43) | -0.81 (-1.08, -0.54) | **<0.001** | | |
| 6-months | 2.53 (1.04) | 2.32 (0.62) | 0.13 (-0.14, 0.40) | 0.36 | | |
| Baseline (ref) | 2.33 (0.86) | 2.25 (0.69) | | | | |
| Participatory decision-making | | | | | 7.43 | **0.02** |
| 12-months | 2.52 (0.93) | 2.75 (0.39) | 0.11 (-0.19, 0.41) | 0.47 | | |
| 6-months | 2.11 (0.90) | 2.69 (0.79) | -0.27 (-0.57, 0.03) | 0.08 | | |
| Baseline (ref) | 1.86 (1.04) | 2.19 (0.97) | | | | |
| Satisfaction with care | | | | | 15.04 | **<0.001** |
| 12-months | 3.09 (0.59) | 3.27 (0.33) | -0.29 (-0.46, -0.13) | **<0.001** | | |

*(Continued)*

**Table 4.** (Continued)

| | Intervention | Control | Intervention Effect by Time Point *Arm\*Time* | | Overall Intervention Effect *Arm\*Time* | |
|---|---|---|---|---|---|---|
| | | | **B (95% CI)** | *p* | Wald χ² | *p* |
| 6-months | 3.29 (0.60) | 3.19 (0.40) | -0.01 (-0.18, 0.16) | 0.91 | | |
| Baseline (ref) | 3.38 (0.55) | 3.26 (0.35) | | | | |
| HIV stigma (overall score) | | | | | 18.72 | **<0.001** |
| 12-months | 0.45 (0.64) | 0.71 (0.50) | -0.28 (-0.44, -0.13) | **<0.001** | | |
| 6-months | 0.59 (0.66) | 0.58 (0.55) | 0.03 (-0.14, 0.19) | 0.76 | | |
| Baseline (ref) | 0.60 (0.65) | 0.60 (0.57) | | | | |
| Anticipated HIV stigma | | | | | 23.84 | **<0.001** |
| 12-months | 0.59 (0.88) | 0.90 (0.63) | -0.46 (-0.71, -0.22) | **<0.001** | | |
| 6-months | 0.80 (0.88) | 0.57 (0.73) | 0.12 (-0.14, 0.37) | 0.37 | | |
| Baseline (ref) | 0.79 (0.89) | 0.67 (0.89) | | | | |
| Enacted HIV stigma | | | | | 4.08 | 0.13 |
| 12-months | 0.12 (0.39) | 0.17 (0.47) | -0.14 (-0.29, 0.01) | 0.06 | | |
| 6-months | 0.16 (0.46) | 0.14 (0.42) | -0.06 (-0.21, 0.09) | 0.45 | | |
| Baseline (ref) | 0.21 (0.52) | 0.14 (0.43) | | | | |
| Internalized HIV stigma | | | | | 5.03 | 0.08 |
| 12-months | 0.62 (0.99) | 1.04 (0.61) | -0.22 (-0.45, 0.01) | 0.06 | | |
| 6-months | 0.82 (1.14) | 1.01 (0.95) | 0.02 (-0.27, 0.32) | 0.88 | | |
| Baseline (ref) | 0.78 (1.13) | 0.99 (0.85) | | | | |

**Abbreviations**: Abbreviations: M = mean, SD = standard deviation, B = unstandardized beta, OR=odds ratio, CI = confidence interval. **Notes**: All provider/staff models control for clinic, cadre, and gender; HIV client model adjustments vary by model based on covariates identified (described in Table A in S1 Text); bold indicates statistical significance at p < 0.05

## Thematic Area 1: Overall Competence in and Provision of Gender-Sensitive HIV Care (Knowledge, Attitudes, Self-Efficacy, Skills)

Providers who received the training reported a significant increase in the overall gender-sensitive competence score compared to those that did not; the overall effect was statistically significant (Wald $\chi^2$ = 20.94, p < 0.001) with the difference observed at 6-months (B = 0.24, 95% CI = 0.12, 0.37, p < 0.001), but not maintained at 12-months (B = -0.02, 95% CI = -0.14, 0.10, p = 0.80) (see Fig 2a). As depicted in Figs 2b and 2c, this effect was stronger for men than women at 6-months, and for lay health workers/non-clinical staff compared to certified health workers and counselors, with the effect attenuated but still maintained for lay health workers/staff at 12-months (statistics for interactions are reported in Tables A and B in S1 Text).

The qualitative exit focus group/interviews corroborate the quantitative data's findings on providers' gender-sensitive care competence, and provides insight into specific improvements, as well as gaps, in knowledge, attitudes, self-efficacy, and related skills (presented next and in Table 5).

**Knowledge.** In qualitative focus groups/interviews with training attendees, participants shared that the training had improved their understanding of gender and HIV. Specifically, providers described learning about the difference between gender and sex, how HIV gender disparities tie to broader gender norms, roles, and inequities, and how gendered barriers to HIV care affect women and men. Discussions in the focus groups on gender vs. sex demonstrated that the training helped some providers understand that people's sex may not match their gender identity or roles, with examples of women doing men's work or men that do not conform to masculine roles, and statements that clients should be treated based on their gender identity or non-conformism, as opposed to their sex.

**Table 5. Qualitative themes with select, illustrative quotations from exit focus groups/interviews with HIV care providers/staff who participated in the training, gender-sensitivity provider training intervention pilot trial, Uganda 2021–2023.**

| Thematic Area 1: Overall Competence in and Provision of Gender-Sensitive HIV Care (Knowledge, Attitudes, Self-Efficacy, Skills) |
|---|

**Knowledge.** HIV providers' narratives showcased increased understanding of sex vs. gender, awareness of HIV gender disparities, and how gender norms/equity affect HIV care engagement and disparities in outcomes for HIV clients. Providers commonly gave applied examples of this knowledge, relating gender norms and roles to men and women's barriers to HIV care engagement and communication with providers.

- Quote 1 (Q1): "*Some of the content was old but some was new. Because 'GBV' we have generally been using that term [Chorus laughter]. We have been using it for a very long time, but what I loved, there was that insight on knowing what is important to men and what is important to women. Because you find that to men, employment was very key. As a result, if someone loses their job, they may fail to adhere [to HIV medication]. So, that can affect how you respond to the client, when they aren't suppressing. The same applies to women; there are some things that they consider very important and if those things aren't going well, then their adherence is affected*" (Records/Data Assistant, Man, Age 33, FGD).

**Gaps**

- Applied examples shared by providers centered on men's barriers to care more than women's.
- Some narratives suggested knowledge may have been overgeneralized (e.g., all men are too busy to come to care).

**Attitudes on the importance of providing gender-sensitive HIV care.** HIV providers expressed increased motivation to be gender-sensitive when responding to clients, which was tied to their increased awareness of the link between gender and HIV care engagement. HIV providers also expressed increased motivation to be responsive to GBV-affected clients through established MOH protocols, which was not viewed as a priority by all providers before the training.

- Q2: "*We were thinking that the challenges of our clients in taking drugs [ARVs] were not connected to GBV. We were concentrating on these others [concerns], like taking drugs. Yet, you may find that she [client] was not taking [ARVs] because the father may be mistreating the child. So, [GBV] screening was missing. It was an eye opener that, fine, we had done the work, but we had to do more. There was something missing!*" (Counselor, Man, Age 34, FGD).

**Self-efficacy and skills to provide gender-sensitive HIV care and shared examples of the use of these skills in client interactions.** Providers expressed increased confidence and gained skills in providing gender-sensitive HIV care, which they said they had put into practice with clients.

- **Communication with men. Q3:** "*It [the training] opened my eyes about the differences in communication between men and women. For example, before I used to find men rude in their communication, but I realized that is them, I think, being the 'leaders.' It's not being rude, but you may attend to a man and when you are communicating, it's as if he is talking to his wife at home. Before, it used to bother me a lot. After the training, I realized that you can actually deal with them, despite that fact*" (Lay Health Worker, Woman, Age 38, FGD).

- **Increased self-efficacy/skills to follow GBV protocols and counsel GBV-affected clients.** Q4: People came with GBV-related challenges and we weren't helping them that much, but, after this training, I realized that I could help these people. They come and tell you that, 'at home it's like this and that. My husband beats me. I am in this situation!' I am now able to counsel them to see that they get helped (Health Worker, Woman, Age 31, Interview).

- **Linking clients to appropriate gender-specific services, like support groups, home visits, male champions, extended hours.** Q5: "*At this time [after the training], if I get a man who is lost [from care], I will send a male champion and I tell him that in such and such a place there is a man who got lost from us. I may not be able to talk to him better than you, so go there to him. If it is a lady, it might be easier for me to go there*" (Lay Health Worker, Woman, Age 34, FGD).

- **Using counseling techniques to help identify areas where HIV is viewed as threatening to gender norms/roles and providing gender-sensitive counseling to reduce the effect of HIV stigma.** Q6: "What attracted the men [to HIV testing since the training] is that we tell them… if you come to the health center, it's not like you have come to the ART clinic to test and they [the community] have found that you are HIV positive. No one at the health center will know about it; there is privacy and you will sit alone in a room with the counsellor and no one will see you. We talked about eliminating fear" (Lay Health Worker, Woman, Age 21, Interview).

**Gaps**

- Participants did not provide many applied examples of using gender-transformative counseling approaches discussed in the training in their practice (aimed at reshaping unhealthy gender norm endorsement).
- There were a few examples of participants reinforcing gender power imbalances or double standards in client interactions (see examples below).
  - Q7: *Even if the wife is the one working and provides everything on the table, you have to take him [the husband] as a man. Yes, you are working, but norms and culture take the man to be the superior. Now, if you are working and getting more money and you go home and disrespect him, he will beat you because he is a man. I got the information and the knowledge on what to tell specific groups. When you are talking to ladies, you tell them…what their responsibility is and how to avoid those GBV cases at home* (Counselor, Woman, Age 29, FGD).
  - Q8: *I got to understand that, while giving services, men appreciate services in different ways compared to the women. So, while giving services, I counsel according to the sex I am dealing with. For instance, I may tell a woman to remain faithful to one partner, which may be not easy to tell the men. So, as I discuss with [men], I let them know that it's not good, but I don't force it on them* (Lay Health Worker, Man, Age 25, FGD).

*(Continued)*

| Thematic Area 2: Overall Competence in and Provision of Client-Centered Care and Communication |
|---|

**Attitudes on the importance of using a client-centered approach in HIV care.** Providers discussed how the training increased their perceived importance of providing client-centered care to clients, noting experienced improvements to their client relationships when incorporating these techniques to their care.

**Self-efficacy and skills to provide client-centered care and shared examples of the use of these skills in HIV client interactions.** Providers discussed gaining skills in client-centered communication and applying these skills in their HIV care. Below are select examples that showcase reoccurring client-centered care components.

- **Building rapport, perspective taking and showing clients empathy, reduced harsh language.** Q9: "Before the training I used to be...call it, harsh, mostly to the clients who would miss their appointments. I wasn't giving them time to explain. But after the training, I realized that these people go through a lot. They face a lot of challenges. So, when she [client] misses [an appointment]. I will come, greet, create rapport and ask, 'I was expecting you on such a date, why didn't you turn up?' Then she gets to explain to me why she didn't make it. But, before…[chorus laughter]" (Health worker, Woman, Age 40, FGD).
- **Giving clients more time, listening to their needs and challenges, and being more equitable.** Q10: "When you handle someone as they are, you will help them. We learned that many times, we think, we, the health workers, know more than the client, so the time you spend with the client you do most of the talking. Yet, they also have brains and think. If you don't give them chance to talk then you may not understand them. Therefore, we give these people [time] and they tell us what they are going through. We have a client-centered discussion to help them based on their story/experiences" (Counselor, Woman, Age 33, FGD).
- **Verbal and non-verbal communication skills.** Q11: "*For the men, I came to understand that they respond more with the non-verbal than the verbal. He can respond with the eyes, head. But the woman, she will keep communicating with you [verbally]. For the men, you have to be very attentive and [use] active listening. After the study, I understood that even the non-verbal [queues] are very key in the sessions*" (Lay Health Worker, Woman, Age 31, FGD).
- **Individualized over group care. Q12:** "*We used to give general talks. You would talk to [clients] and teach every one of them [together]. But now, we also encourage them to come one on one – individual sessions*" (Counselor, Man, Age 34, FGD).
- **Taking a less judgmental approach (discussed further in Theme 3).** Q13: "*When people come here…and tell us, 'they raped me at night.' We used to ask them, 'Why were you walking at night as a woman?' But now, we don't judge them. They come when they have raped them, we don't judge them because we are here to help them. First, help them and see how they get out of this challenge they are facing*" (Counselor, Woman, Age 33, FGD).

**Gaps**
- The implementation of client-centered approaches was challenged by heavy work load, a lack of time, and privacy in the clinic environment.
  - Q14: "*You have to take time with the patient, listen to them, all their stories and baggage, and give them time to express themselves…I don't think we have that environment here. When you have a long line of clients, you aren't going to give everyone 30 minutes [to] 1 hour listening to all their problems and challenges*" (Health worker, Man, Age 26, FGD).

| Thematic Area 3: Bias Recognition, Reduction, and Use of Emotional/Stress Regulation Techniques |
|---|

**Bias recognition.** Providers shared how the training helped them recognize biases they held towards different client groups including: clients who drink (mainly men), sex workers, young women assumed to be promiscuous, men in general considered to be difficult clients, gender and sexual minority individuals, drug users, young people referred to locally as "muyaye",[1] and virally unsuppressed clients.

**Shared examples of providers intentionally using gender-sensitive, client-centered approaches learned in the training with client groups they held bias against.** Providers recognizing their own biases against clients shared examples of intentional efforts to build relationships with clients, empathize with their specific challenges, reduce judgment towards clients, improve their overall treatment of them, and use emotional and stress regulation techniques to prevent stress and prejudice from entering provider-client interactions (see select examples below).

- **HIV clients who drink (mainly men).** Q15: "*After the training, we would sit with [clients who drink alcohol] and tell them to reduce on the alcohol consumed but before that we would tell them, 'First, stay there, far; don't come close.' But now we tell them, 'Reduce on the amount you consume, first take medication' and you tell them about the dangers of alcohol and medication. But before, we would quarrel with them*" (Health worker, Woman, Age 42, FGD).
- **Sex workers**. Q16: "Aha! That time I had no time for those who wear minis [mini-skirts] - I would give them their drugs and I would not care whether you take it properly or not or even throw it away. I am not concerned because I had it in me that, 'you are a prostitute and it is up to you.' But now I take time to interact. When there are things that are not right, we go for counseling sessions, and when there are things which have gone well, we continue to move with the good thing. I now create time and I have time for such people" (Health Worker, Woman, Age 29, FGD).
- **Use of stress management techniques.** Q17: "*We would carry our problems from home and take them to the health facility. By the time the client comes, you are like, 'You should talk properly, because even us, we have our own problems'…So, when we were learning gender, I learnt how to manage stress. You come [to work] with a mood and when they ask you, 'How many clients you have worked on?' – there is not a single one that you have served. You would look at her [a client] and tell her to come back on Wednesday [laughs]. 'You have no one to take care of you, go back!' You have your stress and you add it and impose it on other people*" (Counselor, Woman, Age 29, FGD).

*(Continued)*

**Table 5.** (Continued)

**Gaps**

- Some narratives suggested that the training helped providers recognize stereotypes and try to not make assumptions based on stereotypes, but that it might not have changed all prejudice underlying the stereotypes.
  - Q18: "*Before this training, I had this feeling that whoever came to the clinic indecently dressed and reacts [to an HIV test] and turns positive, that 'This one was a commercial sex worker and was selling themselves; that's why they are HIV positive.' But after the study, I got to realize that, at times, its one's personality and how they feel comfortable [dressing], not that they are commercial sex workers.*" (Health worker, Woman, Age 34, FGD).
- Some providers expressed difficulty in fully letting go of bias, and of stress or anger caused by clients.
  - Q19: "*When I try to put myself in their shoes, he drinks and I also drink, the truth is that I have just put that on myself but that's not what I am. Our characters are different.*" (Lay Health Worker, Woman, Age 31, FGD).

**Thematic Area 4: Translation of the Gender-Sensitivity Training's Effect on HIV Client Outcomes**

**Perceived changes in client openness/communication, with subsequent improvements in HIV care engagement.** Changes in providers' actions were perceived by some providers to improve clients' openness in communication through increased trust.

**Perceived changes in client HIV care engagement outcomes.** Some providers expressed that the training had resulted in noted improvements in clients' HIV care engagement outcomes, but others noted these changes were modest and will take more time to come to fruition at a facility-level.

- Q20: *That gender [training] helped me a lot and removed from me what they would call judging people. For example, we have a girl who comes with her tight trousers and, whenever I would see her, I would see a pure 'muyaye' [laughter]. She was not suppressing and I did not have time for her. I would look at her and wonder what I was going to talk with her about. Now, after getting the training, I got time and started talking to that girl…we kept on communicating and, by the end, she had suppressed. I think, before, she would look at me and she could not tell me about her problem. But when she saw that I was caring and I made her my friend she found I was easy to talk to* (Lay Health Worker, Woman, Age 33, FGD).

Abbreviations: ART=Antiretroviral Treatment; ARVs=Antiretrovirals; GBV=Gender-based violence, FGD=Focus Group Discussion; MOH=Ministry of Health

Notes: Health Workers include clinical officers, nurses, and nurse midwives. Lay Health Workers include HIV peer counselors/expert clients, linkage facilitators, and male champions. Counselors are those with a degree in counseling (not lay counselors).

In addition, providers often gave applied examples of the link between gender norms and clients' challenges; common examples included masculinity-related barriers to engaging men (e.g., work responsibility, stigma, difficulty communicating with them), HIV status disclosure issues for women tied to dependence on male partners, and GBV. Quotation 1 (Q1) in Table 5 also showcases the salient theme that providers gained deeper insight into gender beyond GBV; at the start of the program, participants often thought "gender" referred only to "GBV," and used the terms interchangeably, due to a strong emphasis of GBV in HIV programming in Uganda. Many participants shared how the training helped them to understand gender's effect not only on GBV, but also on HIV outcomes.

While gains in knowledge were salient throughout all focus groups and interviews, some discussions more heavily focused on men's gendered barriers to care than women's – this could suggest the intervention affected knowledge about men more than about women. Further, a subset of discussions displayed an overgeneralization or oversimplification of the content learned through statements, such as all men are too busy to come to the clinic, without deeper discussion of the role of gender norms provided or recognition that such statements may not be true about all men.

**Attitudes towards providing gender-sensitive care.** Taken together, providers' narratives demonstrated increased motivation to provide gender-sensitive care to clients. In addition, across all focus groups, providers explicitly expressed increased motivation to better address GBV among HIV clients by following existing MOH GBV protocols (for screening and referral, which the training reinforced) and to better counsel GBV-affected clients. Many providers disclosed that addressing GBV among clients was not previously a priority to them, and a few providers made clear that they were not previously connecting the importance of addressing GBV to their goals in improving HIV care engagement (see Q2 in Table 5).

**Self-efficacy and skills to provide gender-sensitive care.** In focus groups/interviews, providers displayed increased self-efficacy for providing gender-sensitive care and described specific skills that they acquired relevant to gender-sensitive counseling and service provision. A dominant theme, particularly among women lay workers, was that they perceived gains in their confidence and communication skills to respond to men, based on their understanding of how

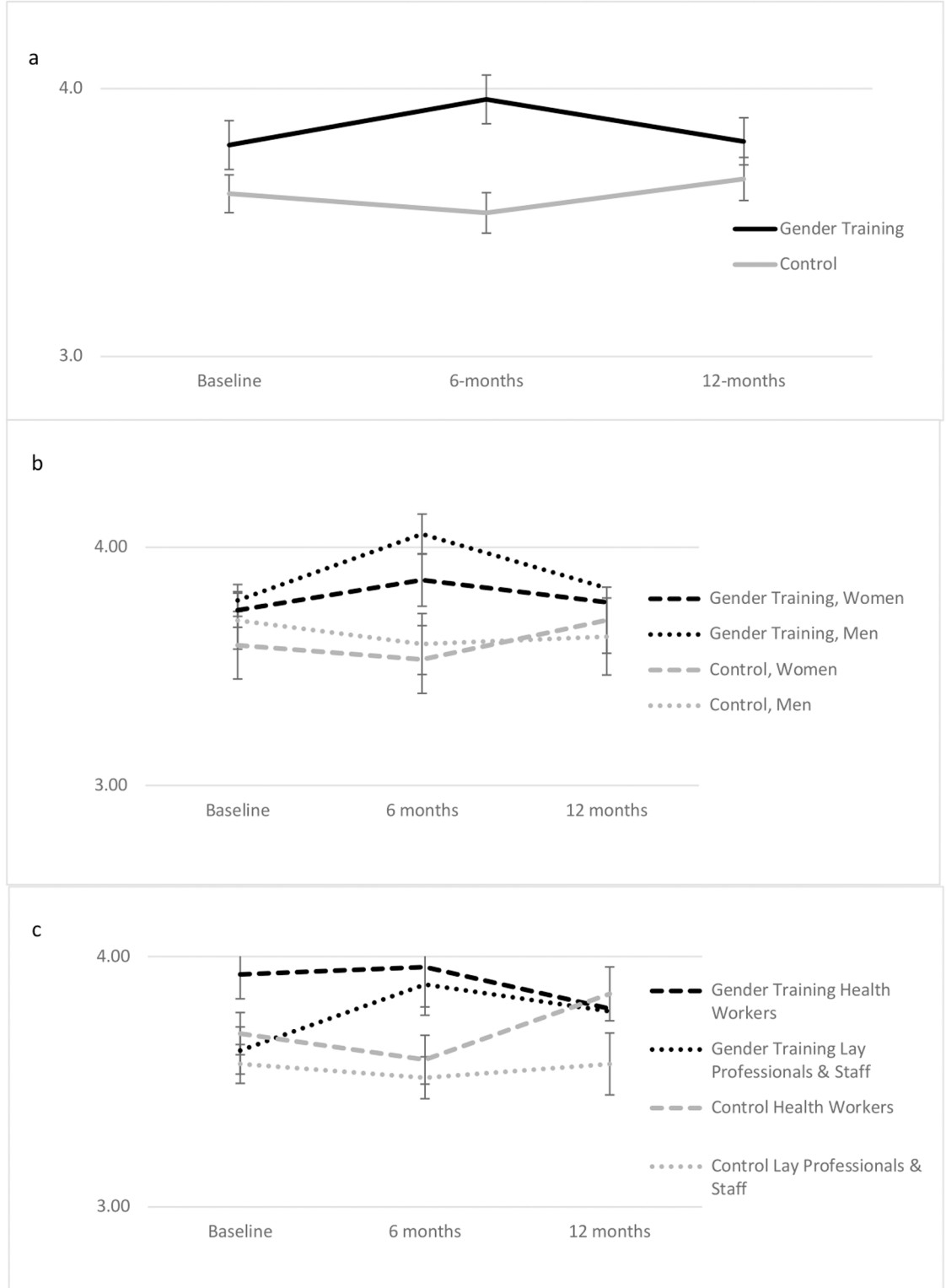

**Fig 2. a-c. Graphic depictions of the intervention by time effect comparing HIV care providers and staff in the gender-sensitivity training intervention and control clinics in competence in gender-sensitive care (primary exploratory outcome) with gender and cadre interactions.** Notes: Models control for clinic, gender, and cadre. Error bars represent 95% Confidence Intervals.

masculine norms affect men's communication style and presentation to care (see Q3 in Table 5). Related to the reported change in attitudes towards responding to GBV, there was also reporting of increased self-efficacy to follow GBV protocols and skills to counsel GBV-affected clients (see Q4 in Table 5).

Providers gave examples that showcased their application of specific skills acquired from the training in their client interactions. Examples mostly fell within gender-sensitive or gender-specific approaches, such as working on men faster who have work conflicts, linking clients to appropriate gender-specific services (e.g., support groups, home visits, male champions), and the use of the "HIV threat framework" to identify and counsel clients on gender-specific stigma (see Q5 and Q6 in Table 5). While the training also discussed gender-transformative counseling approaches aimed to move clients towards gender equitable norms, there were fewer examples of this discussed in practice. Further, there were a few instances of providers sharing examples of counseling patients in ways that maintain traditional gender norms and equity. In examples provided in Table 5 (Q7), one health worker explains how they now coach women on how to not evoke violence from their spouse and another (Q8) explains how they now counsel women on faithfulness as a strategy to reduce HIV transmission, but not men.

## Thematic Area 2: Overall Competence in and Provision of Client-Centered Care and Communication

The training sessions aimed to convey how client-centered care and communication are central to providing gender-sensitive care and strengthen providers' client-centered communication skills. Quantitative data from care providers and clients included several indicators of client-centered care, with mixed results. There was an increase in providers' self-efficacy for client-centered communication in the control clinics; there was a modest increase at 6-months and reduction at 12-months in the intervention providers (Wald $\chi^2$ = 7.84, p = 0.02; 6 months: B = 0.04, 95% CI = -0.18, 0.27, p = 0.70; 12-months: B = -0.22, 95% CI = -0.44, -0.00, p = 0.05). Among clients, perceptions of the quality of communication from HIV care providers followed the same patterns, favoring improvements in the control clinics (Wald $\chi^2$ = 54.80, p < 0.001; 6 months: B = 0.13, 95% CI = -0.14, 0.40, p = 0.36; 12-months: B = -0.81, 95% CI = -1.08, -0.54, p < 0.001). However, providers receiving the training reported increased empathy towards clients compared to control clinic providers at 6-months (Wald $\chi^2$ = 4.26, p = 0.12; 6 months: B = 0.17, 95% CI = 0.00, 0.33, p = 0.04; 12-months: B = 0.13, 95% CI = -0.04, 0.30, p = 0.14). Clients in both arms reported increased perceptions of participatory decision-making with HIV care providers related to their HIV treatment over time (Wald $\chi^2$ = 7.43, p = 0.02; 6 months: B = -0.27, 95% CI = -0.57, 0.03, p = 0.08; 12-months: B = 0.11, 95% CI = -0.19, 0.41, p = 0.47). Further, there was an interaction with sex, with increased participatory decision-making reported among men in the intervention clinics compared to all other groups (i.e., intervention women and control clients) (see Figure A and Table D in S1 Text).

Despite mixed quantitative results, increased recognition of the importance of a client-centered approach, as well as shared experiences integrating this approach into practice, were among the most salient qualitative themes. Providers' most shared examples involved building rapport, giving clients more time, trying to understand their needs and challenges, perspective taking and showing clients empathy, and reducing harsh language and taking a less judgmental approach. A subset of providers also discussed increased recognition of clients' rights to have a say in their treatment and increased individualized care approaches over a "one size fits all" approach. Illustrative quotations that highlight these client-centered elements are in Table 5 (Q9-Q13). Providers explained that these concepts were not new to them but that the intervention reinforced them and helped them put them into practice. They described these efforts as a considerable departure from the typical interactions with clients before the intervention, which were characterized as harsh and lacking empathy (see Q9, Table 5).

While providers' narratives conveyed strong support for uptake of client-centered care, it also provided insight into challenges in implementing client-centered care into practice, mainly related to the clinic environment and workload (i.e., lack of privacy, limited time) (see Q14 in Table 5). Providers also expressed how adopting these practices requires building relationships with clients and trust, which takes time, making progress incremental.

## Thematic Area 3: Bias recognition, reduction, and use of emotional/stress regulation techniques

The gender-sensitivity training included content intended to help HIV care providers understand that all people are affected by bias and stereotypes, and bias against clients living with HIV can unconsciously, negatively affect their provision of equitable, quality HIV care. The content specifically highlighted HIV bias/stereotypes, in general, and as they intersect with gender bias, while also using guided discussions to elicit locally relevant biases. Given the contextually specific nature of bias, change in bias was not quantitatively assessed. However, in exit focus groups and interviews, providers shared how the training helped them recognize biases they held against different groups. Bias discussed centered on people who drink alcohol or use drugs, sex workers, young women assumed to be promiscuous, men in general who were considered difficult clients, gender and sexual minority individuals, "dirty" or unhygienic clients, and virally unsuppressed clients, and young people referred to as "Bayayes." "Muyaye," the singular form of the plural, Bayayes,. is a derogatory term that narrowly refers to a thief or trickster but has evolved to be used more broadly to refer to a style, a manner of speech and dress associated with urban youth in Uganda with lifestyle and livelihoods that do not conform to traditional social norms [55]. These biases were expressed in all focus groups (including health workers and lay health workers). Differences were mainly apparent by clinic, depending on client populations served – for example, providers from clinics serving populations with heavier alcohol use or larger sex worker populations discussed those specific groups more.

There were ample narratives to suggest that, after recognizing their own bias through the bias-specific content, providers used the patient-centered counseling and communication skills acquired from the training to change how they interact with these groups, such as being less judgmental in their approach, displaying empathy, building relationships, giving them adequate time, seeking to understand their problems more deeply, and generally reducing harsh and stigmatizing treatment towards clients (see Q15-Q16 for select examples in Table 5).

In addition, the training taught the use of emotional and stress regulation techniques as a way to prevent bias from entering client interactions. The reported use of these techniques increased in intervention providers compared to control providers over time overall (Wald $\chi^2$=8.17; p=0.01) and at 6-months (6 months: B=0.42, 95% CI=0.13, 0.72, p=0.005; 12-months: B=0.30, 95% CI=-0.01, 0.61, p=0.05). Participants also discussed using these techniques in the exit interviews (see Q18 in Table 5).

However, there were narratives that suggested that, while the training helped providers recognize stereotypes and consciously try to avoid making assumptions about individual clients based on them, underlying prejudice remained. An example is provided in Table 5 (Q18); in this case, a lay health worker expresses how they learned that all women who wear mini-skirts are not sex workers. Some providers' narratives demonstrated increased understanding and empathy to broader contextual factors that lead women to engage in sex work – but, in this case, there is no evidence that the underlying prejudice towards sex workers changed for this lay health worker. Further, providers were forthright in the challenges that exist in overcoming bias and in using emotional/stress regulation techniques to let go of stress or anger held towards specific clients (see Q19 in Table 5).

## Thematic Area 4: Translation of the gender-sensitivity training's effect on outcomes for clients living with HIV

In the client cohort followed over 12-months, ART adherence (main outcome) improved over time in both study arms with no difference between the two (Wald $\chi^2$=2.53, p=0.28; 6 months: B=0.39, 95% CI=-6.67, 7.45, p=0.91; 12-months: B=-3.05, 95% CI=-9.21, 3.11, p=0.33). Adherence to HIV clinic appointments followed the same pattern (Wald $\chi^2$=1.68, p=0.43; 6 months: B=-0.01, 95% CI=-0.04, 0.03, p=0.86; 12-months: B=-0.02, 95% CI=-0.05, 0.02, p=0.32). Clients' self-reported satisfaction with HIV care was static in the control clinics overtime with a modest reduction in intervention clinic clients, particularly between 6- and 12-months (Wald $\chi^2$=15.04, p<0.001; 6 months: B=-0.01, 95% CI=0.18, 0.16, p=0.91; 12-months: B=-0.29, 95% CI=-0.46, -0.13, p<0.001). In favor of the intervention, clients in the training clinics reported a reduction in HIV stigma over time compared to clients in the control clinics (where an increase was observed) (Wald $\chi^2$=18.72, p<0.001; 6 months: B=0.03, 95% CI=-0.14, 0.19, p=0.76; 12-months: B=-0.28, 95% CI=-0.44, -0.13, p<0.001). Examining the

subscales that make up the overall HIV stigma score, the intervention arm's reduction in HIV stigma was driven by reduced anticipated HIV stigma (Wald $\chi^2$=23.84, p<0.001; 6 months: B=0.12, 95% CI=-0.14, 0.37, p=0.37; 12-months: B=-0.46, 95% CI=-0.71, -0.22, p<0.001). The other two stigma subscales were also reduced in the intervention compared to control, but not at a level of statistical significance (enacted HIV stigma: Wald $\chi^2$=4.08, p=0.13; 6 months: B=-0.06, 95% CI=-0.21, 0.09, p=0.45; 12-months: B=-0.14, 95% CI=-0.29, 0.01, p=0.06; internalized stigma: Wald $\chi^2$=5.03, p=0.08; 6 months: B=0.02, 95% CI=-0.27, 0.32, p=0.88; 12-months: B=-0.22, 95% CI=-0.45, 0.01, p=0.06).

In providers' qualitative focus groups and interviews, many participants shared that they felt the changes they made in their interactions with clients had resulted in observable, positive effects on client outcomes. Most salient was the perception that improved treatment of clients, especially with those facing barriers to HIV care engagement, had improved patients' openness with providers (see Q20, Table 5). Many providers said that they felt these changes were translating into better client engagement in HIV care or related outcomes, such as alcohol reduction in clients who drink. However, some providers noted that building relationships with clients and seeing the fruits of gender-sensitive, client-centered care takes time and perceived the translation of change into patient outcomes as only modest at this stage.

## Discussion

The findings of this pilot trial provide preliminary support for a training intervention's ability to improve providers' competence for gender-sensitive HIV care in health settings in Uganda. Through a mixed methods evaluation, the findings specifically support the training's potential to improve providers' knowledge, attitudes, and skills to respond to clients' gendered barriers to HIV care engagement. The findings also support the promise of using of a gender norms lens as an appropriate framework to increase client-centered care and reduce gender and other related biases towards clients. In this pilot, the observed provider effects largely did not translate into clients' quantitatively captured outcomes related to HIV care engagement, perceived quality of provider communication, or satisfaction with care. However, providers reported improvements in their relationships with clients, and in turn, perceived a positive effect of the intervention on client outcomes. Further, the client cohort data demonstrated an increase in men's perception of their level of participatory decision-making in their HIV care and a reduction in HIV stigma among clients in the intervention clinics. As a pilot trial, these effects are preliminary and need to be replicated in a larger, fully powered trial.

This pilot trial found preliminary support that participation in this training increased providers' competence to provide gender-sensitive care. This was evidenced through modest, greater relative change in intervention vs. control participants in our overall competence measure. The measure was inclusive of providers' knowledge of HIV gender disparities, perceived importance of providing gender-sensitive care, and self-efficacy to provide gender-sensitive care. The qualitative data validated and expanded on these findings—showcasing increased understanding of gender vs. sex, how gender affects men and women's communication and care engagement, and the importance of considering gender norms in providing HIV services and counseling to clients. Notably, the intervention's effect on this outcome was attenuated by 12-months, highlighting the potential need for more follow-up support. In addition, the effect was larger for men vs. women providers and for lay health workers and staff vs. certified health workers and counselors. These groups may have had less previous training on the topic, but there may also be a need to consider tailoring the content by group in the future. The qualitative data also highlighted a potentially positive effect of the intervention on responding to GBV—with narratives to support increased perceived importance of following GBV protocols and skills to handle GBV clients. Thus, this gender-sensitivity training could help overcome known barriers to implementation of existing GBV protocols in HIV clinical settings in Uganda and similar settings [25], but in the future testing of this intervention, quantitative metrics should be added to validate these findings.

The training intervention included content to help providers strengthen their client-centered communication skills and use them to help identify and address clients' gendered barriers to HIV care. Providers who attended the training qualitatively reported efforts to build rapport with clients, reduce harsh language and judgement, and use perspective taking to

put themselves in clients' shoes. Through quantitative measures, there was also an intervention effect on increased empathy towards clients—which others have linked to better treatment of patients living with HIV [56]. We anticipated improved client-centered care might also address some previously identified barriers to HIV care engagement that intersect with gender in provider-client relationships (inequitable power dynamics, gendered communication styles, masculine norms of respect conflicting with harsh treatment from providers) [9,10]. For men who received the intervention, there was a sustained intervention effect on the perceived level of participatory decision-making between clients and providers. This may have been an outcome of this aspect of the intervention, which may have been most relevant for improving care with men, who providers (mostly women) discussed having difficulty communicating with prior to the intervention.

In our study, however, other quantitative indicators of client-centered care did not have an effect or improved in the control more than in the intervention. As a pilot trial, it may be that the study needed more clusters, a larger sample size, and more time to see these results fully translate into client outcomes. Providers did note that relationship-building takes time. Moreover, the study highlighted considerable barriers to the provision of client-centered care related to the clinic environment (time/space constraints). Others have pointed to these same barriers as limiting the provision of client-centered care throughout sub-Saharan Africa, as well as a dearth of research on how to improve client-centered care in the region [32]. A recent pilot trial in Kenya showed the promise of research with this aim—reporting increased viral suppression among clients living with HIV in clinics that received a patient-centered care intervention [57]. Considering the challenges to scaling up client-centered care in this and similar settings, adding more than one practicum-based follow-up in the future scale up of this intervention may help support the translation of content into practice in facilities. In addition, future research focused on gender-sensitive capacity building could also explore ways to formally change institutional procedures, practices, or environmental contexts in ways to support gender-sensitive, client-centered care provision.

This pilot also supports the potential positive effects of pairing client-centered care promotion with bias reduction training, grounded in a gender norms lens. The qualitative findings highlighted increased empathy towards different groups of clients that providers held gender bias towards. Among the general population, which was the training's focus, negative attitudes were expressed towards men in general, non-suppressed clients, young women, youth generally, and clients who drink alcohol. Similar to others [26], a salient finding was providers' expressed frustration and difficulty communicating with men, who were perceived as rude and impatient. Our findings suggest that teaching providers about how gender norms affect men's care engagement can increase their empathy towards men and build skills to counsel them. Future research and programming aimed to address the "HIV blind spot" in men's care engagement should consider these findings and the role HIV care providers can play in either insolating or engaging men living with HIV.

Although this study focused on cis-gender heterosexual women and men, introducing gender as a social construct provided an entry point for discussions on gender and sexual minorities, and intersections with other stigmatized groups. As a result, after the training, providers expressed increased empathy towards, and reported intentional efforts to be client-centered with, key populations, including gender and sexual minority individuals and sex workers. Other intervention studies in Uganda have shown that sensitivity trainings with providers can improve their attitudes and empathy for key populations [35,58]. Using a broad gender-norms lens for sensitivity trainings can be explored as a potentially culturally acceptable way to increase provider empathy for and reduce bias against all clients. This has important implications in a climate that may not currently support sensitivity trainings explicitly directed towards these groups.

Increased client-centered communication and reduced bias towards clients may explain this pilot study's observed reduction of HIV stigma reported among clients in the intervention arm clinics vs. those in the control arm clinics. Providers characterized their communication with clients before the training as harsh and lacking empathy, especially towards those struggling with treatment adherence. Providers also described stigmatizing treatment of some groups, such as isolating men and those who drink, reprimanding young girls and sex workers for the way they dress, and, in some cases, victim-blaming rape victims. Such treatment is likely to increase a clients' experience of HIV stigma, which is associated with poor care engagement [59,60].

The training may have also reduced HIV stigma among clients through the content that explained clients' experience of HIV stigma through a gender norms lens. Facilitators discussed men and women's gender norms and roles, and illustrated how HIV can be viewed as threatening to those ideals. Facilitators taught providers counseling techniques to elicit clients' "HIV threats" and mitigate HIV stigma by: 1) counseling them away from norms that do not align with HIV care engagement (gender-transformative), 2) highlighting ART's ability to help them restore/maintain their roles (gender-specific), and 3) linking them to services that could reduce HIV stigma (e.g., male champions, support groups) (gender-sensitive/specific). While providers' qualitative narratives included applied examples of the latter two, a potential gap was highlighted in the focus groups through relatively less discussion of the use of gender-transformative counseling. Moreover, there were a few examples of providers citing the training on gender norms but using it to uphold traditional norms that underpin gender inequity between men and women. Others have warned about similar unintentional consequences of gender-focused public health work [61]. Taken together, these findings support the continued exploration of interventions that address HIV stigma reduction through a gender norms lens, but underscore the need for intensive training and oversight in the implementation of gender-sensitive and transformative interventions to ensure all messaging is received as intended.

This pilot trial is not powered to fully evaluate this intervention's effects. While these data support this intervention's promise, the findings should be replicated and expanded on in a future, fully powered trial. In addition to the findings presented here, this trial collected mixed methods data to examine acceptability, feasibility, and factors affecting implementation [39], the findings of which will inform the refinement of this intervention before future testing. Beyond the sample size limitations of a pilot trial, the pilot had few clusters, limiting our ability to fully account for within-cluster correlation in our analysis. Cluster randomization was necessary to reduce the risk of contamination between arms and to establish procedures for a future multisite trial. However, having only two clusters introduces the risk of confounding variables, and we did identify differences between clients in the two arms at baseline. Clients in the intervention clinic had higher reporting of several factors known to negatively affect HIV care engagement, including intimate partner violence history, alcohol use, and food insecurity. While we controlled for baseline differences in analyses, these differences may have reduced the likelihood of seeing an intervention effect in this group relative to the control. We also saw changes in the control arm on different outcomes, which could be explained by measurement reactivity or external factors, such as other programs or trainings, which cannot be ruled out with this pilot design. Given the strong covariate balance in the provider cohort and the small sample size, we did not apply inverse probability of treatment weighting, though we acknowledge it as an alternative approach in larger studies with greater imbalance. Another limitation of the study is the use of mainly self-reported measures. In a future hybrid effectiveness-implementation trial, the study will collect viral load directly from clients. Consequently, participants' responses may have been influenced by recall and social desirability bias.

The mixed methods design of this study is a strength. The two data sources served to validate each other, and the qualitative data provided insight into providers' experience with the intervention and into nuanced findings that are difficult to capture quantitatively, such as the findings on provider bias. However, the qualitative data are also subject to social desirability bias. While the exit focus groups were conducted by a moderator independent of study team, providers may have still felt the need to provide positive feedback on the intervention. A future trial could strengthen the mixed methods approach to further validate or contest these findings through observational methods, supervisor interviews to corroboration or challenge reported effects, and the collection of quantitative process measures that could capture some of the changes described qualitatively (such total number of clients screened for GBV).

## Conclusions

This training intervention responds to calls to modify health systems, and HIV care specifically, to be more gender-responsive. Providers and staff with regular interaction with clients can play a critical role in reducing gender norm-related barriers to HIV care for men and women. This pilot trial provides preliminary support for a training intervention

that improves gender-sensitive care by increasing providers' awareness of gender norms and HIV gender disparities, building empathy and reducing gender and other bias towards clients, and strengthening client-centered communication skills to respond to clients' gender-based preferences and barriers to HIV care engagement. This intervention should be fully tested in a future, hybrid effectiveness-implementation trial to establish its effects on provider and client outcomes. If found efficacious, this intervention would have important public health implications for Uganda and potentially for other settings where gender norms influence HIV outcomes. Future research could also explore the adaption of this approach for improving the provision of health services beyond HIV where providers' delivery of services and clients' outcomes are similarly affected by gender norms.

## Supporting information

**S1 Text. Supplemental files including sub-analyses of intervention effects (figures and tables) and CONSORT checklist.**
(DOCX)

**S1. Data. Syntax for all analyses using client data in SPSS.**
(SPS)

**S2. Data. Client data in long format, inclusive of client cohort data used for intervention effect analyses in SPSS.**
(SAV)

**S3. Data. Client data in long format, inclusive of client cohort data used for intervention effect analyses in Excel.**
(XLSX)

**S4. Data. Client data in a wide format, inclusive of client cohort data used for descriptive analyses and analyses of baseline equivalence in SPSS.**
(SAV)

**S5. Data. Client data in a wide format, inclusive of client cohort data used for descriptive analyses and analyses of baseline equivalence in Excel.**
(XLSX)

**S6.Data. Syntax for all analyses using provider data in SPSS.**
(SPS)

**S7. Data. Provider data in long format, inclusive of provider cohort data used for intervention effect analyses in SPSS.**
(SAV)

**S8. Data. Provider data in long format, inclusive of provider cohort data used for intervention effect analyses in Excel.**
(XLSX)

**S9. Data. Provider data in a wide format, inclusive of provider cohort data used for descriptive analyses and analyses of baseline equivalence in SPSS.**
(SAV)

**S10. Data. Provider data in a wide format, inclusive of provider cohort data used for descriptive analyses and analyses of baseline equivalence in Excel.**
(XLSX)

## Author contributions

**Conceptualization:** Katelyn M. Sileo, Rhoda K. Wanyenze, Sten H. Vermund, Shari L. Dworkin, John F. Dovidio, Trace S. Kershaw.

**Data curation:** Katelyn M. Sileo, Rhoda K. Wanyenze, Asha Anecho, Barbara Mukasa, Semei C. Mukama, Trace S. Kershaw.

**Formal analysis:** Katelyn M. Sileo, Asha Anecho, Rebecca L. Luttinen, Katherine Weston.

**Funding acquisition:** Katelyn M. Sileo, Trace S. Kershaw.

**Investigation:** Katelyn M. Sileo, Rhoda K. Wanyenze, Trace S. Kershaw.

**Methodology:** Katelyn M. Sileo, Rhoda K. Wanyenze, Barbara Mukasa, Sten H. Vermund, Shari L. Dworkin, John F. Dovidio, Barbara S. Taylor, Trace S. Kershaw.

**Project administration:** Katelyn M. Sileo, Rhoda K. Wanyenze, Asha Anecho, Semei C. Mukama.

**Resources:** Katelyn M. Sileo.

**Software:** Katelyn M. Sileo.

**Supervision:** Katelyn M. Sileo, Rhoda K. Wanyenze, Sten H. Vermund, Shari L. Dworkin, John F. Dovidio, Barbara S. Taylor, Trace S. Kershaw.

**Validation:** Katelyn M. Sileo.

**Visualization:** Katelyn M. Sileo.

**Writing – original draft:** Katelyn M. Sileo.

**Writing – review & editing:** Rhoda K. Wanyenze, Asha Anecho, Rebecca L. Luttinen, Katherine Weston, Barbara Mukasa, Semei C. Mukama, Sten H. Vermund, Shari L. Dworkin, John F. Dovidio, Barbara S. Taylor, Trace S. Kershaw.

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
