## [Decision Letter · Decision Letter 0]

12 Feb 2025

PGPH-D-24-02900

Mixed methods pilot evaluation of a gender-sensitivity training for HIV care providers in Uganda: Effects on providers and clients

Dear Dr. Sileo,

Thank you for submitting your manuscript to PLOS Global Public Health. After careful consideration, we feel that it has merit but does not fully meet PLOS Global Public Health’s publication criteria as it currently stands. Therefore, we invite you to submit a revised version of the manuscript that addresses the points raised during the review process.

We look forward to receiving your revised manuscript.

Kind regards,

Guillaume Fontaine, PhD, RN

Academic Editor

Journal Requirements:

2. Please provide an Author Summary. This should appear in your manuscript between the Abstract (if applicable) and the Introduction, and should be 150–200 words long. The aim should be to make your findings accessible to a wide audience that includes both scientists and non-scientists. Sample summaries can be found on our website under Submission Guidelines:

https://journals.plos.org/globalpublichealth/s/submission-guidelines#loc-parts-of-a-submission.

Additional Editor Comments (if provided):

Reviewers' comments:

Reviewer's Responses to Questions

**Comments to the Author**

1. Does this manuscript meet PLOS Global Public Health’s publication criteria?

Reviewer #1: Partly

Reviewer #2: Partly

2. Has the statistical analysis been performed appropriately and rigorously?

Reviewer #1: Yes

Reviewer #2: No

3. Have the authors made all data underlying the findings in their manuscript fully available (please refer to the Data Availability Statement at the start of the manuscript PDF file)?

Reviewer #1: Yes

Reviewer #2: Yes

4. Is the manuscript presented in an intelligible fashion and written in standard English?

Reviewer #1: No

Reviewer #2: Yes

Reviewer #1: The nature of the study is quite interesting and relevant being a pilot study and issue of public health importance. It is also of note that pilot studies usually face challenges especially in terms of methodology and communication of findings. The authors did fairly well in overcoming the challenges. However, to make the manuscript worthy of publishing in line with standards of PLOS Public Health policies, the following weaknesses were noticed and should be addressed.

1. The authors are advised to study well the citation style of PLOS Public Health and adhere strictly, particularly in-text citations.

2. The manuscript seem voluminous, hence, contains some sentences considered unnecessary and needless repetitions e.g. in Page 4, line 3 and 4, one sentence could have represented the two sentences on selected facilities.

3. I could not find any mention of ethical permission which I consider necessary with the nature of the study.

4. Make sampling techniques and methodology in concise and familiar statements to make them reproducible for other intending researchers.

5. There was no mention of what was done to minimize bias in the selection of respondents and this can lead to cofounders and/or chance.

6. Findings should be made concise and point-blank. Kindly report findings on quantitative and qualitative study separately.

7. Improve in the use of punctuation in the entire manuscript e.g in Page 4, line 3 and 4, one study found health workers had explicit, negative....

Reviewer #2: General comments:

This manuscript presents the results from an gender-sensitivity training intervention for HIV care providers. The study was performed in six clinics in Uganda.

To set expectations, I am a quantitative researcher, so I will reserve comment on the qualitative

aspects of this manuscript, mostly on the statistical methods. My read is that the statistical analyses focus around the use of regression models that utilize generalized estimating equations (GEE) to account for clinic-level clustering and repeated measures by participant.

First, regarding the statistical power, clustering was not taken into account. For a group randomized trial (GRT), Murray et al (Am J Public Health, 2004;94:423) commented that, "...it is unusual for a GRT to have adequate power with fewer than 8 to 10 groups per condition."

Second, GEE-based models are often referred to as population-averaged or marginal models. As such, they generally need quite a few clusters to ensure that the GEE approach is able to provide a valid marginal estimate. In the "Data analysis approach" subsection it is not stated what is used as the clustering variable. My assumption is that you have treated clinical as the cluster. In that case, there are far too few clusters for a marginal model. The Murray et al. article also quotes another article of their suggesting that GEE/sandwich estimator standard errors require 40+ total groups (20+ per arm in a balanced study). There are some small number of clusters corrections (e.g., Mancl-DeRouen is one I remember), but I am not sure if these are available in SPSS. Leyrat et al. (Intl J Epidemiol, 2018;47:321) concludes that the corrections are very important to correct the type I error rate, but are very underpowered.

Thus, my major comment about your analyses is that I think this study is likely underpowered even with corrections. That is something that goes against criterion #3 in the PLOS ONE publication criteria (https://journals.plos.org/plosone/s/criteria-for-publication). I know that you have indicated in your language that this is a "pilot trial" and the results are "preliminary" and provide "promise". All of that concurs with the nature of this trial, but the quantitative methods chosen (especially GEE) do not concur with this.

The most common alternative is to run a conditional model, usually a mixed model also known as a random effects (RE) model. The interpretation differs (it is at the individual level rather than the group level) but mixed models require fewer clusters because the results are at an individual level rather than a group level. Though, given Murray et al.'s comment about GRTs, you are also in a situation where you don't really have enough clusters for robust comparisons between arms. That's because the random intercepts per clinic are used to build a distribution, but it's hard to build a distribution with so few intercepts.

Unfortunately, I think the analyses fall into that situation where you have too few clusters for REs and GEE, but enough that it feels necessary to account for clustering. I thought there was a paper that suggested including the clusters as fixed effects provides (roughly) the same coefficients and SEs as treating the clusters as random effects in a situation like this, but I am not able to find it at this time. Maybe I'm thinking of Abadie et al. (Quar J Economics, 2023;138:1. Regardless, that is my recommendation here is to include the clusters as fixed effects in a regression model and note in the limitation that there are not enough clusters to account for within-cluster correlation. Though, I am open to other suggestions.

Another question I had is whether the intervention is given in small groups. In that case, you might have an individually randomized group treatment trial (IGRT). The key here is whether the intervention is delivered in small groups. I suggest reading this article https://ajph.aphapublications.org/doi/full/10.2105/AJPH.2007.127027 to see if your design would qualify as an IGRT and adjust accordingly. I bring this up because the small groups may be a more accurate representation of the dependency between observations in these data. Also, it would give you more clusters and maybe GEE will work then and I've written this for nothing.

Specific comments:

1. (p.8) What is a "cell"?

2. Including line numbers greatly helps with the review process and I encourage you to submit papers with line numbers in the future.

**Do you want your identity to be public for this peer review?** For information about this choice, including consent withdrawal, please see our Privacy Policy

Reviewer #1: **Yes: ** Sunday Charles Adeyemo

Reviewer #2: No

---

## [Decision Letter · Decision Letter 1]

18 Jun 2025

PGPH-D-24-02900R1

Mixed methods pilot evaluation of a gender-sensitivity training for HIV care providers in Uganda: Effects on providers and clients

Dear Dr. Sileo,

Thank you for submitting your manuscript to PLOS Global Public Health. After careful consideration, we feel that it has merit but does not fully meet PLOS Global Public Health’s publication criteria as it currently stands. Therefore, we invite you to submit a revised version of the manuscript that addresses the points raised during the review process.

The manuscript has been evaluated by the two original reviewers, and their comments are available below.

Reviewer 1 has some requests for minor edits, and reviewer 2 has a number of comments and questions regarding the analyses.

Could you please carefully revise the manuscript to address all comments raised?

We look forward to receiving your revised manuscript.

Kind regards,

Steve Zimmerman, PhD

PLOS Staff Editor

Journal Requirements:

Additional Editor Comments (if provided):

Reviewers' comments:

Reviewer's Responses to Questions

**Comments to the Author**

Reviewer #1: All comments have been addressed

Reviewer #2: (No Response)

publication criteria?

Reviewer #1: Yes

Reviewer #2: Partly

3. Has the statistical analysis been performed appropriately and rigorously?

Reviewer #1: Yes

Reviewer #2: No

4. Have the authors made all data underlying the findings in their manuscript fully available (please refer to the Data Availability Statement at the start of the manuscript PDF file)?

Reviewer #1: Yes

Reviewer #2: Yes

5. Is the manuscript presented in an intelligible fashion and written in standard English?

Reviewer #1: Yes

Reviewer #2: Yes

Reviewer #1: The Authors have demonstrated desires to improve on their manuscripts as all issues raised in the previous revision have been addressed, I want to advise the Authors to make some minor improvements in the manuscript.

1) conjunction “And” appears to have been overused in line 85 of the manuscript

2)I’m of the view that statement in line 90 can be better written as “However there is a gap in researches that focus on building healthcare ……..” except if the Authors strongly believe otherwise.

Reviewer #2: General comments:

This manuscript reports the results of an intervention focused on gender-sensitive HIV training implemented in Uganda using a mixed methods approach.

First off, I struggled with the cause and effect in this trial. Although randomization is performed at the clinic-level and the clinics are matched, the analyses are all at the patient level. Thus, there is no guarantee that the randomization will balance the intervention and control groups at the individual level. Indeed, you mention this in the limitations, but then there is nothing really done to address this, other than throwing variables in a regression model and hoping it works out. This is not the typical design since there is some randomization, and multiple sources of variability (clinic clusters, maybe the intervention group, longitudinal measurements). But I think more could be done to achieve a better balance between groups to provide a stronger causal effect. Inverse probability of treatment weighting might be an option.

Also, it seemed as though the intervention was applied in small groups. I couldn't tell if people stayed in the same group. If so, that could be another source of variability since there could be correlation between members of the same group.

I also wondered about how useful this study is a pilot, based on the limitations mentioned in lines 703-706. You mentioned that an efficacy trial will collect viral load from patients and not use self-report measures. Your primary aim (line 142) seems to be to gather preliminary evidence. Not collecting data with the outcome that would be used in the trial seems like a big shortcoming. Do you have information on how reliable self-report data are?

I had many comments on analytic aspects of the manuscript. Those are below in the specific comments section.

Specific comments:

1. (lines 43,52 and throughout the manuscript) I'd recommend shifting away from test statistics and p-values and reporting the point estimate and confidence interval (CI). That conveys more information to readers. In addition, it concurs with the shift away from p-values.

2. (lines 46-47, 353-354, and throughout the manuscript) Same theme as my previous comment; it's more useful to have the CI rather than a standard error and p-value.

3. (line 212) Citation needed for G * Power. Always include the version of any software used.

4. (lines 205-216) I was confused by the statement in lines 206-208 that is quoting 15-30 per condition and then the rest of the paragraph which computes a power of 100 per condition. What point is being made by having the statement in lines 206-208?

5. (line 339) I don't know what are "generalized linear equations" are. Do you mean generalized linear models? I strongly, strongly recommend to always have methodological citations for your statistical methods. That way if this terminology is domain-specific (or misused) then a reader has a much better chance of determining the methods used.

6. (lines 341-342) I know that people like to say they used GEE in methods sections, but unfortunately GEE is an optimization method and not a statistical method. GEE could be used along with many different regression-type methods. I presume you have used logistic regression with GEE to account for repeated measures. Please confirm this and include this information in your revisions.

7. (lines 343-344) I don't disagree with what you have done by treating clinic as a fixed effect, though I don't understand what is being said here. It's impossible to include both random effects and GEE in the same model since they do not use the same optimization method. Given that is impossible, what are you arguing? Random effects are generally better than GEE-based approaches when there are a small number of clusters since there is good evidence that the sandwich estimator that is used in GEE is biased when the number of clusters is less than 40 (see Emrich and Piedmonte, J Stat Computation Simulation. 1992;41:19, Lipsitz et al, Biometrics. 1994;50:270, Feng et al, Stat Med. 1996;15:1793, Murray et al, Eval Rev. 1996;20:313). That bias is downward, meaning the confidence intervals are too small. Thus, a random effects model would be a better option with a small number of clusters. Are you arguing against a multilevel model with patients (and their longitudinal measurements) nested within clinics? Usually random effects models start having a chance to work when there are >=6 clusters, which is what you have. It's also important to note that the interpretations are different so it's not a direct swap between the two methods, but it's close.

8. (Lines 345-347) In regards to variable selection, this approach is sometimes referred to as "bivariate screening". This approach has been criticized in literature (see Harrell, Jr. _Regression Modeling Strategies_. New York: Springer; 2001. and Wiegand, _Statist Med_, 2010;29:1647 https://doi.org/10.1002/sim.3943) and an example has been shown to miss important variables (Sun et al., _J Clin Epidemiol_, 1996;49:907 http://dx.doi.org/10.1016/0895-4356\(96\)00025-X). Regardless, the results from one of those studies (Wiegand, _Statist Med_, 2010;29:1647) suggested that at sample sizes of 1,000 or greater, stepwise variable selection procedures may do adequately well, especially when used with a larger p-value than 0.05. Sun et al. found that bivariate screening can miss a variable that may be a confounder even when a p-value higher than what you have used. Bivariate screening can be considered a form of stepwise variable selection, which usually does not do a good job of finding the most appropriate model (e.g., [https://doi.org/10.1002/sim.3943](https://doi.org/10.1002/sim.3943)). Generally it's better to select based on more robust criteria. I prefer shrinkage-based models, such as lasso, for variable selection. At a minimum, I would encourage using measures which assess the fit of the model. AIC or BIC may be possible, for which you could you Burnham and Anderson (doi: 10.1177/0049124104268644) as a guide.

9. (Line 688) In this line you talk about effectiveness but then later in the paragraph you mention doing a community efficacy trial. Which is it? Efficacy is the effect of an intervention under ideal circumstances and effectiveness is the effect of an intervention in real-world practice.

10. (Line 696) Why are you saying there are only two clusters? I thought there were six clinics and those were the clusters.

**Do you want your identity to be public for this peer review?** For information about this choice, including consent withdrawal, please see our Privacy Policy

Reviewer #1: **Yes: ** Sunday Charles Adeyemo

Reviewer #2: No

---

## [Decision Letter · Decision Letter 2]

19 Aug 2025

PGPH-D-24-02900R2

Mixed methods pilot evaluation of a gender-sensitivity training for HIV care providers in Uganda: Effects on providers and clients

Dear Dr. Sileo,

Thank you for submitting your manuscript to PLOS Global Public Health. After careful consideration, we feel that it has merit but does not fully meet PLOS Global Public Health’s publication criteria as it currently stands. The revisions provided have addressed the reviewers' comments in full. We note below a few minor copy-editing issues, and one concern regarding the over-interpretation of p values. Especially given the large number of results reported and the use of two time points for each outcome, it would be preferable to focus the text on reporting of the actual magnitude and 95% CI of effects, removing confusing language on 'trends / trending', and simply stating as needed when results showed statistical significance at p < 0.05.

We look forward to receiving your revised manuscript.

Kind regards,

Hannah Hogan Leslie, PhD

Academic Editor

Journal Requirements:

Additional Editor Comments:

Results paragraph on baseline imbalance - should that be ‘treatment arms differed at baseline (p<0.05)’ not p<0.50?

Please consider providing the units for the Beta values in this paragraph - years for HIV status and ART use, thousands of shillings for income? Hours for travel time?

Above Table 2- extraneous ‘)’

Results Thematic Area 2, next page ‘clients in both treatment arms’ == ‘clients in both arms’ since there is 1 intervention and 1 control arm?

Thematic area 3 quantitative results on emotional and stress regulation - ‘simple’ is non-specific, but presumably means unadjusted or crude? In addition, p values do not trend, they simply are or are not less than a given threshold. Please revise to state the results directly, e.g. self-reported use of regulation techniques was higher in the intervention than control arm at 6 months (B=0.42, …) and 12 months (B=0.30….)

Similarly in Thematic Area 4 in the paragraph on stigma, the phrasing around trends introduces confusion as to whether the statistical test is an actual test of trends vs. the analysis as described that compares 6 month to baseline and 12 month to baseline. Preferable to report the betas and CIs and note if needed the effects with p<0.05

Reviewers' comments:

Reviewer's Responses to Questions

**Comments to the Author**

Reviewer #1: All comments have been addressed

Reviewer #2: All comments have been addressed

publication criteria?

Reviewer #1: Yes

Reviewer #2: (No Response)

3. Has the statistical analysis been performed appropriately and rigorously?

Reviewer #1: Yes

Reviewer #2: (No Response)

4. Have the authors made all data underlying the findings in their manuscript fully available (please refer to the Data Availability Statement at the start of the manuscript PDF file)?

Reviewer #1: Yes

Reviewer #2: (No Response)

5. Is the manuscript presented in an intelligible fashion and written in standard English?

Reviewer #1: Yes

Reviewer #2: (No Response)

Reviewer #1: The Authors have addressed all the previous raised , even though the manuscript seems to be voluminous however all the information seem to be relevant obviously a part of challenges associated with pilot

studies.

Reviewer #2: (No Response)

**Do you want your identity to be public for this peer review?** For information about this choice, including consent withdrawal, please see our Privacy Policy

Reviewer #1: **Yes: ** Sunday Charles Adeyemo

Reviewer #2: No

---

## [Editor Report · Decision Letter 3]

31 Aug 2025

Mixed methods pilot evaluation of a gender-sensitivity training for HIV care providers in Uganda: Effects on providers and clients

PGPH-D-24-02900R3

Dear Dr Sileo,

We are pleased to inform you that your manuscript 'Mixed methods pilot evaluation of a gender-sensitivity training for HIV care providers in Uganda: Effects on providers and clients' has been provisionally accepted for publication in PLOS Global Public Health.

Best regards,

Hannah Hogan Leslie, PhD

Academic Editor